# The quest for the GRAph Level autoEncoder (GRALE)

**Paul Krzakala**
LTCI & CMAP , Télécom Paris, IP Paris

**Gabriel Melo**
LTCI, Télécom Paris, IP Paris

**Charlotte Laclau**
LTCI, Télécom Paris, IP Paris

**Florence d'Alché-Buc**
LTCI, Télécom Paris, IP Paris

**Rémi Flamary**
CMAP, Ecole Polytechnique, IP Paris

## Abstract

Although graph-based learning has attracted a lot of attention, graph representation learning is still a challenging task whose resolution may impact key application fields such as chemistry or biology. To this end, we introduce GRALE, a novel graph autoencoder that encodes and decodes graphs of varying sizes into a shared embedding space. GRALE is trained using an Optimal Transport-inspired loss that compares the original and reconstructed graphs and leverages a differentiable node matching module, which is trained jointly with the encoder and decoder. The proposed attention-based architecture relies on Evoformer, the core component of AlphaFold, which we extend to support both graph encoding and decoding. We show, in numerical experiments on simulated and molecular data, that GRALE enables a highly general form of pre-training, applicable to a wide range of downstream tasks, from classification and regression to more complex tasks such as graph interpolation, editing, matching, and prediction.[1]

## 1 Introduction

**Graph representation learning.** Graph-structured data are omnipresent in a wide variety of fields ranging from social sciences to chemistry, which has always motivated an intense interest in graph learning. Machine learning tasks related to graphs are mostly divided into two categories: node-level tasks such as node classification/regression, clustering, or edge prediction, and graph-level tasks such as graph classification/regression or graph generation/prediction [7, 38, 69, 28, 84]. This paper is devoted to graph-level representation learning, that is, unsupervised learning of a graph-level Euclidean representation that can be used directly or fine-tuned for later tasks [25]. From the large spectrum of existing representation learning methods, we focus on the AutoEncoder approach, as it natively features the possibility to decode a graph back from the embedding space. In this work, we illustrate that this enables leveraging the learned representation in a variety of downstream tasks, ranging from classification/regression to more involved tasks such as graph interpolation, editing, matching or prediction.

**From node-level AutoEncoders...** Scrutinizing the literature, we observe that most existing works on graph AutoEncoders provide node-level embeddings instead of one unique graph-level embedding (See Fig. 1, left). In the following, we refer to this class of models as Node-Level AutoEncoders. The most emblematic example is the celebrated VGAE model [36], where the encoder is a graph

---

[1]Code available at `https://github.com/KrzakalaPaul/GRALE`

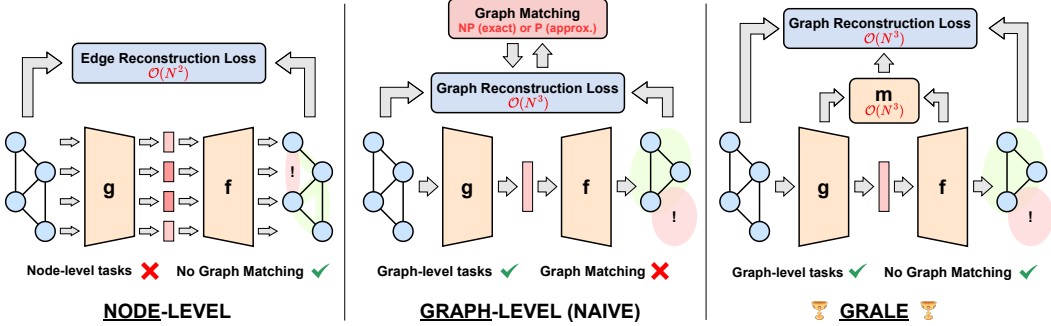

Figure 1: The different classes of graph AutoEncoders. (Left) Node-level AutoEncoders such as [36] provide node level embeddings. (Center) Naive graph-level AutoEncoders such as [53] directly provide graph-level embeddings but rely on a graph matching solver to compute the training loss. (Right) Matching free approaches, such as proposed in this work and in [68] use a learnable module to provide the matching.

convolutional network (GCN) that returns node embeddings $z_i$, and the decoder reconstructs the adjacency matrix from a dot product $A_{i,j} = \sigma(\langle z_i, z_j \rangle)$. Many extensions have been proposed for this model, such as adversarial regularization [48] or masking [56], and it has been shown to be efficient for many node-level tasks, such as node clustering [81, 82, 60] or node outlier detection [14]. Node-level models can also be used for graph generation, given that one knows the size (number of nodes) of the graph to generate; this includes GVAE [36] but also GraphGAN [61] and graph normalizing flows [44]. When a graph-level representation is needed, one can apply some pooling operation on the node embeddings, for instance $z = \sum z_i$. Yet, this strategy is not entirely satisfying, as it inevitably leads to information loss, and it becomes difficult to decode a graph back from this representation [59, 68].

**...to graph-level AutoEncoders.** In contrast, very few works attempt to build graph-level AutoEncoders where the encoder embeds the full graph into a single vector and the decoder reconstructs the whole graph (including the number of nodes). The Graph Deconvolutional AutoEncoder [41] and Graph U-net [21] are close to this goal, but in both cases, the decoder takes advantage of the ground-truth information, including the graph size, in the decoding phase. Ultimately, we identify only two works that share a similar goal with this paper: GraphVAE [53] (not to be confused with GVAE [36]) and PIGVAE [68]. GraphVAE is a pioneering work, but it is heavily based on an expensive graph matching algorithm. In that regard, the main contribution of PIGVAE is the addition of a learnable module that predicts the matching instead. This was a major step forward, yet we argue that PIGVAE is still missing some key components: for instance, the ground truth size of the graph needs to be provided to the decoder. We detail our positioning with respect to PIGVAE in Section 4.

**Contributions.** We introduce a GRAph Level autoEncoder (GRALE) that encodes and decodes graphs of varying sizes into a shared Euclidean space. GRALE is trained with an Optimal Transport-inspired loss, without relying on expensive solvers, as the matching is provided by a learnable module trained end-to-end with the rest of the model. The proposed architecture leverages the Evoformer module [30] for encoding and introduces a novel "Evoformer Decoder" for graph reconstruction. GRALE enables general pretraining, applicable to a wide range of downstream tasks, including classification, regression, graph interpolation, editing, matching, and prediction. We demonstrate these capabilities through experiments on both synthetic benchmarks and large-scale molecular datasets.

## 2 Building a Graph-Level AutoEncoder

### 2.1 Problem Statement

The goal of this paper is to learn a graph-level AutoEncoder. Given an unsupervised graph dataset $\mathcal{D} = \{x_i\}_{i \in 1,...,n}$ we aim at learning an encoder $g : \mathcal{G} \mapsto \mathcal{Z}$ and a decoder $f : \mathcal{Z} \mapsto \mathcal{G}$ where $\mathcal{Z}$ is an euclidean space and $\mathcal{G}$ is the space of graphs. To this end, the classic AutoEncoder approach is to minimize a reconstruction loss $\mathcal{L}(f \circ g(x_i), x_i)$. However, the Graph setting poses unique challenges compared to more classical AutoEncoders (e.g., images):

1. The encoder $g$ must be permutation invariant.

2. The decoder $f$ must be able to map vectors of fixed dimension to graphs of various sizes.
3. The loss $\mathcal{L}$ must be permutation invariant, differentiable and efficient to compute.

**Permutation invariant encoder.** The first challenge is a well-studied topic for which a variety of architectures have been proposed, most of which rely on message passing [83, 84, 73] or attention-based [47, 50] mechanisms. In this work, we have chosen to use an attention-based architecture, the Evoformer module from AlphaFold [30]. As the numerical experiments show it, this architecture is particularly powerful, enabling the encoding and updating of pairwise relationships between graph nodes. To maintain symmetry with the encoder, the decoder also uses a novel *Evoformer Decoder* module. Details are provided in Appendix C.

**Padded graphs for multiple output sizes.** The second challenge can be mitigated by using a classical padded representation [37, 52]. We represent a graph $G \in \mathcal{G}$ as a triplet $G = (h, F, C)$ where $h \in [0,1]^N$ is a masking vector, $F \in \mathbb{R}^{N \times n_f}$ is a node feature matrix, and $C \in \mathbb{R}^{N \times N \times n_c}$ is an edge feature tensor. $N$ is a maximum graph size; all graphs are padded to this size. A node $i$ exists if $h_i = 1$ and is a padding node if $h_i = 0$. Thus, original and reconstructed graphs of various sizes are represented with fixed-dimensional tensors that can be efficiently parametrized.

**Permutation invariant loss.** The third challenge is arguably the hardest to overcome, since any permutation-invariant loss $\mathcal{L}_{\text{PI}}$ between graphs $G$ and $\hat{G}$ can be written as a matching problem

$$\mathcal{L}_{\text{PI}}(G, \hat{G}) = \min_{P \in \sigma_N} \mathcal{L}_{\text{ALIGN}}(G, P[\hat{G}]), \tag{1}$$

where $\sigma_N$ is the set of permutation matrices and $P[G]$ denote the application of permutation $P$ to graph $G$ i.e. $P[G] = (Ph, PF, PCP^T)$. This can be seen as first, aligning the two graphs, and then computing a loss $\mathcal{L}_{\text{ALIGN}}$ between them. This is problematic because, for any nontrivial loss $\mathcal{L}_{\text{ALIGN}}$ such that $\mathcal{L}_{\text{ALIGN}}(x, y) = 0 \iff x = y$, the optimization problem in (1) is NP complete [20]. We discuss how to avoid this pitfall in the next section.

## 2.2 Matching-free reconstruction loss

A common approach to mitigate the computational complexity of a loss like (1) is to relax the discrete optimization problem into a more tractable one. For instance, Any2Graph [37] relaxes the graph matching problem into an Optimal Transport problem $\mathcal{L}_{\text{A2G}}(G, \hat{G}) = \min_{T \in \pi_N} \mathcal{L}_{\text{OT}}(G, \hat{G}, T)$ optimized over $\pi_N = \{T \in [0,1]^{N \times N} \mid T\mathbf{1}_N = \mathbf{1}_N, T^T\mathbf{1}_N = \mathbf{1}_N\}$ the set of bi-stochastic matrices and

$$\mathcal{L}_{\text{OT}}(G, \hat{G}, T) = \sum_{i,j}^{N} \ell_h(h_i, \hat{h}_j)T_{i,j} + \sum_{i,j}^{N} h_i \ell_F(F_i, \hat{F}_j)T_{i,j} + \sum_{i,j,k,l}^{N} h_i h_k \ell_C(C_{i,k}, \hat{C}_{j,l})T_{i,j}T_{k,l}, \tag{2}$$

where $G = (h, F, C)$, $\hat{G} = (\hat{h}, \hat{F}, \hat{C})$, and $\ell_h, \ell_F, \ell_C$ are ground losses responsible for the correct prediction of node masking, node features and edge features respectively. Despite the relaxation, this loss still satisfies key properties as stated in Propositions 1 and 2 (proofs in Appendix F).

**Proposition 1.** *If $\ell_C$ is a Bregman divergence, then $\mathcal{L}_{OT}(G, \hat{G}, T)$ can be computed in $\mathcal{O}(N^3)$.*

**Proposition 2.** *There exist $T \in \pi_N$ such that $\mathcal{L}_{OT}(G, \hat{G}, T) = 0$ if and only if there exist $P \in \sigma_N$ such that $G = P[\hat{G}]$ (i.e. $G$ and $\hat{G}$ are isomorphic).*

Unfortunately, the inner optimization problem w.r.t. $T$ is still NP-complete, as it is quadratic but not convex. The authors suggest an approximate solution using conditional gradient descent, with a complexity of $\mathcal{O}(k(N)N^3)$, where $k(N)$ is the number of iterations until convergence. However, there is no guarantee that the optimizer reaches a global optimum.

Another approach, first proposed in PIGVAE [68], is to completely bypass the inner optimization problem by **making the matching $T$ a prediction of the model**, that is, the model does not only reconstruct a graph $\hat{G} = f \circ g(G)$, but it also provides the matching $\hat{T}$ between output and input graphs.

We propose to combine the loss (2) from Any2Graph with the matching prediction strategy from PIGVAE. Thus, the loss that we minimize to train GRALE is

$$\mathcal{L}_{\text{GRALE}}(G) = \mathcal{L}_{\text{OT}}\left(G, f \circ g(G), \hat{T}(G)\right). \tag{3}$$

Note that $\hat{T}(G)$ must be differentiable so that the model can learn $\hat{T}$, $f$ and $g$ by backpropagation.

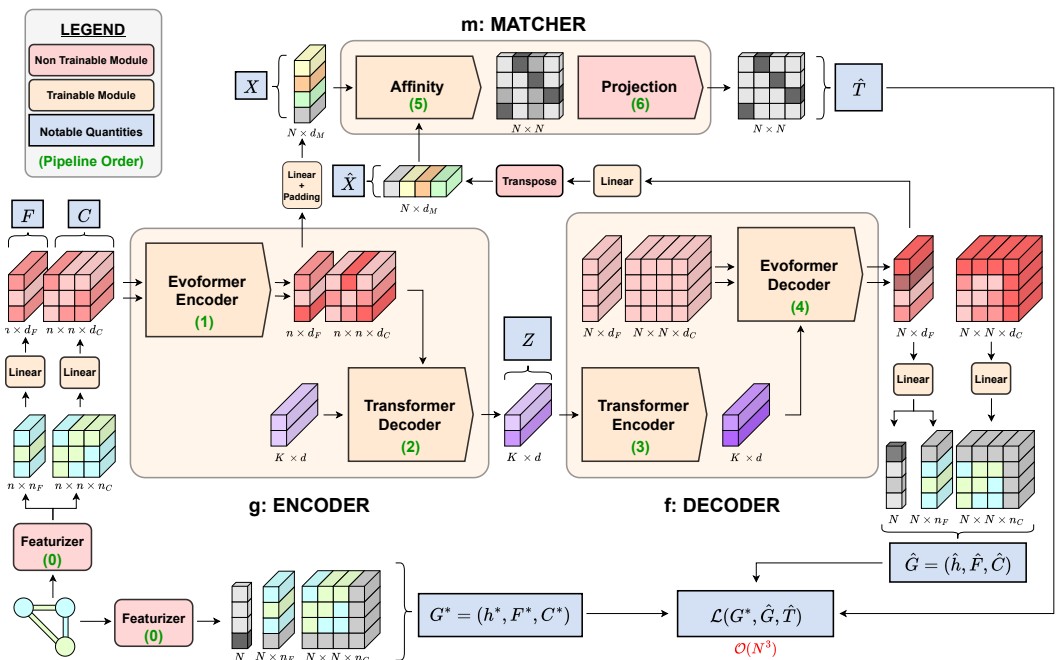

Figure 2: GRALE illustrated for an input of size $n = 3$ and a maximum output graph size $N = 4$.

# 3 GRALE architecture

## 3.1 Overview

The design of the GRALE architecture revolves around the parametrization of the matching matrix $\hat{T}$ between the input graph $G$ and the reconstructed graph $\hat{G}$. Our approach follows a classical idea: approximating the intractable Graph Matching problem (NP-hard) via linear assignment (cubic complexity) of some learnable node embeddings [? 63, 49, 42]. Concretely, we extract node embeddings $g_{\text{nodes}}(G) = X$ and $f_{\text{nodes}}(\hat{G}) = \hat{X}$, and compute $\hat{T} = \text{MATCHING}(X, \hat{X})$, where MATCHING denotes a differentiable matching algorithm detailed below. A key feature of our design is that $g_{\text{nodes}}$ and $f_{\text{nodes}}$ are not treated as independent modules; instead, we employ extensive weight sharing so that these components reuse the same hidden representations as the encoder and decoder.

We next provide the details of this implementation.

## 3.2 Graph pre-processing and featurizer

The first step is to build the input (resp. target) graph $G$ (resp. $G^*$) from the datapoint $x \in \mathcal{D}$

$$\phi(x) = G, \quad \phi^*(x) = G^* \tag{4}$$

This module first extract from data[2] a simple graph presentation with adjacency matrix and node labels then builds additional node and edge features, by including higher order interactions such as the shortest path matrix. Note that we consider different schemes for the input and target graphs ($\phi \neq \phi^*$) as it has been shown that breaking the symmetry between inputs and targets can be beneficial to AutoEncoders [74]. In particular, $\phi$ outputs slightly noisy node features while $\phi^*$ is deterministic. This is crucial for breaking symmetries in the input graph, enabling the encoder to produce distinct node embeddings, which in turn facilitates matching. We discuss this phenomenon in appendix B.3.

## 3.3 Encoder $g$

The input graph $G = (F, C)$ is passed through the encoder $g$ that returns **both** a graph level embedding $Z \in \mathbb{R}^{K \times D}$ and node level embeddings $X \in \mathbb{R}^{N \times d_n}$.

$$g_{\text{graph}}(G) = Z, \quad g_{\text{nodes}}(G) = X \tag{5}$$

---

[2]For molecules, $x$ is typically a SMILES string that can be converted into a graph using RDKit [39].

**Embedding.** We represent the graph embedding as a set of $K$ tokens, each of dimension $D$, with both $K$ and $D$ fixed. For most downstream tasks, we simply flatten this representation into a single vector of dimension $d = K \times D$. This token-based approach follows a broader trend of modeling data as sets of abstract units (tokens) beyond NLP [29, 55, 26]. In this context, $K$ can be interpreted as the number of concepts used to represent the graph, though a qualitative analysis is beyond the scope of this paper. We discuss the choice of $K$ and $D$ quantitatively in Section 5.

**Architecture.** The main component of the encoder is an *Evoformer* module that provides hidden representation for the node feature matrix $F$ and edge feature tensor $C$. Then, we use a *transformer decoder* as a pooling function to produce the graph level representation $Z$ and apply a simple linear layer on $F$ to get the node level representations $X$.

### 3.4 Decoder $f$

The decoder uses the graph-level embedding $Z$ to reconstruct a graph $\hat{G}$ and its node embeddings $\hat{X}$ that will be used for matching as discussed in the next subsection.

$$f_{\text{graph}}(Z) = \hat{G}, \quad f_{\text{nodes}}(Z) = \hat{X} \tag{6}$$

The decoder architecture mirrors that of the encoder. First, a *transformer encoder* updates the graph representation $Z$, then a novel *Evoformer decoder* module reconstructs the output graph nodes and edges. This new module, detailed in Appendix C, is based on the original Evoformer module, augmented with cross-attention blocks to enable decoding. As before, $\hat{X}$ is obtained by applying a simple linear layer to the last layer hidden node representations.

### 3.5 Matcher $m$ and loss

Finally, a matcher module leverages the node embeddings $X$ and $\hat{X}$ to predict the matching between input and output graphs:

$$\hat{T} = m(X, \hat{X}) \tag{7}$$

To enforce that $\hat{T} \in \pi_N$, we decompose the matcher in two steps: 1) We construct an affinity matrix $K$ and $K_{i,j} = \texttt{Aff}(X_i, \hat{X}_j)$ where $\texttt{Aff}$ is a learnable affinity function 2) We apply the Sinkhorn algorithm to project $K$ on $\pi_N$ in a differentiable manner i.e. $\hat{T} = \texttt{SINKHORN}(K)$ [16, 19, 23].

**GRALE loss.** Omitting the preprocessing and denoting by $\phi, \psi$ and $\theta$ the parameters of the encoder, decoder, and matcher respectively, the expression of the loss function writes as follows:

$$\mathcal{L}_{\text{GRALE}}(G, \phi, \psi, \theta) = \mathcal{L}_{\text{OT}}\left(G, \underbrace{f^\phi_{\text{graph}} \circ g^\psi_{\text{graph}}(G)}_{\hat{G}}, \underbrace{m^\theta\left(g^\psi_{\text{nodes}}(G), f^\phi_{\text{nodes}} \circ g^\psi_{\text{graph}}(G)\right)}_{\hat{T}}\right). \tag{8}$$

The detailed implementation of the modules mentioned in this section is provided in Appendix C and the whole model is trained end-to-end with classic batch gradient descent as described in A.

## 4 Related Works

### 4.1 Permutation Invariant Graph Variationnal AutoEncoder (PIGVAE)

We devote this section to highlighting the differences with the work of Winter et al. [68] who first proposed a graph-level AutoEncoder with a matching free loss (PIGVAE).

**Graph size.** In PIGVAE, the decoder needs to be given the ground truth size of the output graph and the encoder is not trained to encode this critical information in the graph-level representation. In comparison, GRALE is trained to predict the size of the graph through the padding vector $h$. In the following, we assume that graphs are of fixed size ($h = \hat{h} = \mathbf{1}$) to make the methods comparable.

**Loss.** Denoting $\hat{T}[\hat{G}] = (\hat{T}\hat{F}, \hat{T}\hat{C}\hat{T}^T)$ the reordering of a graph, PIGVAE loss rewrites as

$$\mathcal{L}_{\text{PIGVAE}}(G, \hat{G}, \hat{T}) = \mathcal{L}_{\text{ALIGN}}(G, \hat{T}[\hat{G}]) \tag{9}$$

where $\mathcal{L}_{\text{ALIGN}}(G, \hat{G}) = \sum_{i=1}^{N} \ell_F(F_i, \hat{F}_i) + \sum_{i,j=1}^{N} \ell_C(C_{i,j}, \hat{C}_{i,j})$. This reveals that $\mathcal{L}_{\text{PIGVAE}}$ and $\mathcal{L}_{\text{OT}}$ can be seen as two different relaxations of the same underlying matching problem $\min_{P \in \sigma_N} \mathcal{L}_{\text{ALIGN}}(G, P[\hat{G}])$. We detail this relationship in Proposition 3. However, Proposition 4 highlights an important limitation of the relaxation chosen in PIGVAE as the loss can be zero without a perfect graph reconstruction. All proofs are provided in appendix F.

**Proposition 3.** *For a permutation $P \in \sigma_N$, $\mathcal{L}_{PIGVAE}$ and $\mathcal{L}_{OT}$ are equivalent to a matching loss,*

$$\mathcal{L}_{PIGVAE}(G, \hat{G}, P) = \mathcal{L}_{OT}(G, \hat{G}, P) = \mathcal{L}_{ALIGN}(G, P[\hat{G}]) \tag{10}$$

*If we relax to $\hat{T} \in \pi_N$ we only have an inequality instead $\mathcal{L}_{PIGVAE}(G, \hat{G}, \hat{T}) \leq \mathcal{L}_{OT}(G, \hat{G}, \hat{T})$.*

**Proposition 4.** *It can be that $\mathcal{L}_{PIGVAE}(G, \hat{G}, \hat{T}) = 0$ while $\hat{G}$ and $G$ are not isomorphic.*

**Architecture.** PIGVAE architecture is composed of 3 main blocs: encoder, decoder and permuter. The role of the permuter is similar to that of our matcher but it only relies on one-dimensional sorting of the input node embeddings $X$ while we leverage the Sinkhorn algorithm to compute a $d$ dimensional matching between $X$ and $\hat{X}$ which makes our implementation more expressive. Besides, training the permuter requires to schedule a temperature parameter. The scheduling scheme is not detailed in the PIGVAE paper which makes it hard to reproduce the reported performances.

## 4.2 Attention-based architectures for graphs

The success of attention in NLP [58] has motivated many researchers to apply attention-based models to other types of data [32, 67]. For graphs, attention mainly exists in two flavors: node-level and edge-level. In the case of node-level attention, each node can pay attention to the other nodes of the graph, resulting in an $N \times N$ attention matrix that is then biased or masked using structural information [79, 77, 50]. Similarly, edge-level attention results in a $N^2 \times N^2$ attention matrix. To prevent these prohibitive costs, all edge-level attention models use a form of factorization, which typically results in $\mathcal{O}(N^3)$ complexity instead [9, 68, 30]. Since the complexity of our loss and that of our matcher is also cubic[3], it seemed like a reasonable choice to use edge-level attention in our encoder and decoder. To this end, we select the Evoformer module [30], as it elegantly combines node and edge-level attention using intertwined updates of node and edge features. Our main contribution is the design of a novel *Evoformer Decoder* that enables Evoformer to use cross-attention for graph reconstruction.

## 4.3 Graph Matching with Neural Networks

Our model relies on the prediction of the matching between input and output graphs. In that sense, we are part of a larger effort towards approximating graph matching with deep learning models in a data-driven fashion. However, it is important to note that most existing works treat graph matching as a regression problem, where the inputs are pairs of graphs $(G_1, G_2)$ and the targets $y$ are either the ground truth matching [80, 63, 66, 78, 64] or the ground truth matching cost (edit distance) [65, 31, 3, 42, 46], or both [49]. In contrast, we train our matcher without any ground truth by simply minimizing the matching cost, which is an upper bound of the edit distance. Furthermore, our matcher is not a separate model; it is incorporated and trained end-to-end with the rest of the AutoEncoder.

## 5 Numerical experiments

**Training datasets.** We train GRALE on three datasets. First, COLORING is a synthetic graph dataset introduced in [37] where each instance is a connected graph whose node labels are colors that satisfy the four color theorem. Specifically, we train on the COLORING 20 variant, which is composed of 300k graphs of size less than or equal to 20. Then, for molecular representation learning, we download and preprocess molecules from the PUBCHEM database [33]. We denote PUBCHEM 32 and PUBCHEM 16 as the subsets containing 84M and 14M molecules, respectively, with size up to 32 and 16 atoms. PUBCHEM 16 is used for training a lightweight version of GRALE in the ablation studies, while all downstream molecular tasks use the model trained on PUBCHEM 32. We refer to appendix A.1 for more details on the datasets, and A.2 for the training parameters.

Table 1: Reconstruction performances of graph-level AutoEncoders. $^*$ indicates that the decoder relies on the ground truth size of the graph. N.A. indicates that the model is too expensive to train.

| Model | COLORING | | PUBCHEM 16 | | PUBCHEM 32 | |
| --- | --- | --- | --- | --- | --- | --- |
| | Edit. Dist. ($\downarrow$) | GI Acc. ($\uparrow$) | Edit. Dist. ($\downarrow$) | GI Acc. ($\uparrow$) | Edit. Dist. ($\downarrow$) | GI Acc. ($\uparrow$) |
| GraphVAE | 2.13 | 35.90 | 3.72 | 07.8 | N.A. | N.A. |
| PIGVAE$^*$ | 0.09 | 85.30 | 1.69 | 41.0 | 2.53 | 24.91 |
| GRALE | **0.02** | **99.20** | **0.11** | **93.0** | **0.78** | **66.80** |

Table 2: Ablation Studies (PUBCHEM 16). We report (avg $\pm$ std) reconstruction metrics over 5 runs.

| Model component | Proposed | Replaced by | Edit Dist. | GI Acc. |
| --- | --- | --- | --- | --- |
| Loss | $\mathcal{L}_{OT}$ | $\mathcal{L}_{PIGVAE}$ + Regularization [68] | $0.13 \pm 0.16$ | $91.8 \pm 5.01$ |
| Featurizer $\phi$ | Second Order | First Order | $0.24 \pm 0.17$ | $90.3 \pm 6.46$ |
| Encoder $f$ | Evoformer | GNN | $1.32 \pm 0.16$ | $53.2 \pm 2.69$ |
| Decoder $g$ | Evoformer | Transformer Decoder [37] | $0.71 \pm 0.09$ | $66.6 \pm 2.78$ |
| Matcher $m$ | Sinkhorn | Softsort [68] | $1.47 \pm 0.36$ | $49.7 \pm 9.15$ |
| Disambiguation Noise | With | Without | $1.16 \pm 0.03$ | $64.4 \pm 1.64$ |
| GRALE (no remplacement) | | | $\mathbf{0.11 \pm 0.04}$ | $\mathbf{93.0 \pm 0.18}$ |

## 5.1 Reconstruction performances and ablation studies

In this section, we evaluate the performance of AutoEncoders based on the graph reconstruction quality using the graph edit distance (Edit Dist) [62] and Graph Isomorphism Accuracy (GI Acc), i.e., the percentage of graphs perfectly reconstructed (see Appendix A.3 for details on the metrics). All models are trained on PUBCHEM 16, with a holdout set of 10,000 graphs for evaluation.

**Model performance and impact of components.** First, we compare GRALE to the other graph-level AutoEncoders (PIGVAE [68], GraphVAE [53]). Table 1 shows that GRALE outperforms the other models by a large margin. Next, we conduct ablation studies to assess the impact of the individual components of our model. For each component, we replace it with a baseline (detailed in E) and measure the effect on performance. Results are shown in Table 2. Overall, all new components contribute to performance improvements. We also report the variance of the reconstruction metrics over 5 training seeds, revealing that while the choice of loss function and featurizer has limited impact on average performance, it significantly improves training robustness.

**Size of the embedding space.** We now focus on the choice of embedding space and report reconstruction accuracy over a grid of values for $K$ and $D$ (Figure 3). As expected, accuracy increases with the total embedding dimension $d = K \times D$. For a given fixed $d$, the performance is generally better when $K > 1$, with optimal results typically around $K \approx D$. Interestingly, this choice is also computationally favorable as the cost of a transformer encoder scales with $\mathcal{O}(d \max(K, D))$. These findings align with the broader hypothesis that many types of data benefit from being represented as tokens [29, 55, 26], a direction already explored in recent theoretical works [17].

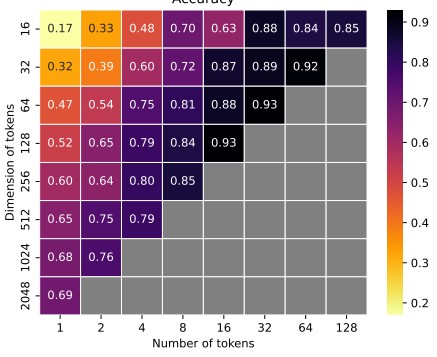

Figure 3: G.I. accuracy vs $(K, D)$. Both axes are in log-scale so that the diagonals correspond to a given total dimension $d = K \times D$.

## 5.2 Qualitative properties of GRALE

**Complex graph operations in the latent space.** We now showcase that GRALE enables complex graph operations with simple vector manipulations in the embedding space. For instance, graph interpolation [6, 18, 27], is traditionnally defined as the Fréchet mean with respect to some graph distance $\mathcal{L}$ i.e. $G_t = \arg\min_G (1 - t)\,\mathcal{L}(G_0, G) + t\,\mathcal{L}(G_1, G)$, which is challenging to compute.

---

$^3$Sinkhorn is $\mathcal{O}(N^2)$, but at test time we use the Hungarian algorithm, its discrete, $\mathcal{O}(N^3)$ counterpart.

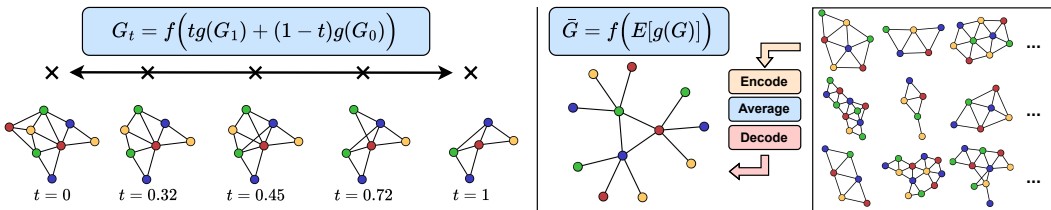

Figure 4: Interpolating graphs (from COLORING) using GRALE's latent space. On the left we interpolate between two graphs while on the right we compute the barycenter $\bar{G}$ of the **whole** dataset.

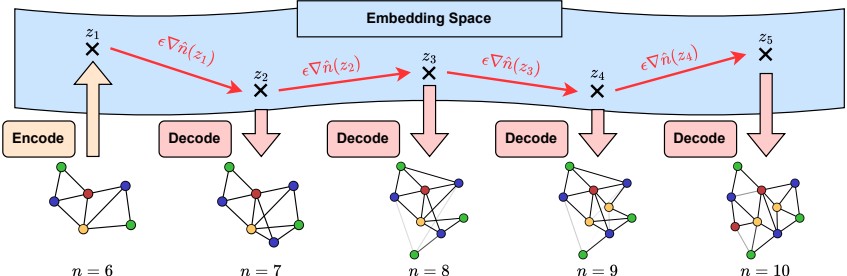

Figure 5: Latent space edition of the size of a graph. Here, $\hat{n}$ is a one-hidden-layer MLP trained to predict graph size, and we set $\epsilon = 0.01$. Steps that did not produce any visible change are omitted.

Instead, we propose to compute interpolations at lightspeed using GRALE's embedding space via $G_t = f((1-t)\,g(G_0) + t\,g(G_1))$. This approach can also be used to compute the barycenter of more than 2 graphs, including an entire dataset as illustrated in Figure 4.

We can also perform property-guided graph editing. Given a property of interest $p(G) \in \mathbb{R}$, we train a predictor $\hat{p} : \mathcal{Z} \to \mathbb{R}$ such that $p(G) \approx \hat{p}(g(G))$, and compute a perturbation $u$ such that $\hat{p}(g(G)+u) \geq \hat{p}(g(G))$. For instance, one can set $u = \epsilon \nabla \hat{p}(g(G))$. The edited graph is then decoded as $G' = f(g(G) + u)$. In Figure 5, we illustrate this on COLORING by setting $p(G) = n(G)$, the size of the graph, and successfully increasing it.

**Denoising Properties.** We explore GRALE's denoising capabilities by corrupting the COLORING test set through random modifications of node colors, potentially creating invalid graphs where adjacent nodes share the same color. We then plot, as a function of noise level, the probability that a noisy graph is valid versus that its reconstruction is valid. As shown in Figure 6, reconstruction significantly increases the proportion of valid graphs, highlighting that GRALE's latent space effectively captures the underlying data structure.

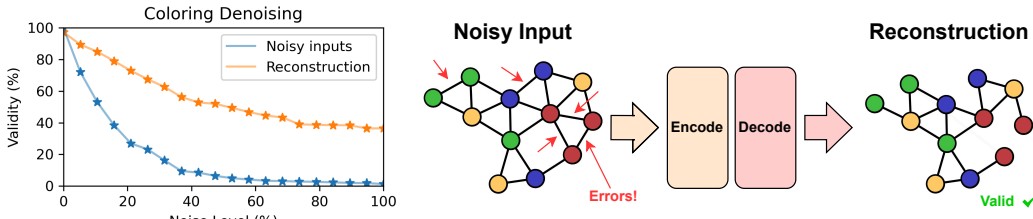

Figure 6: Left: Percentage of valid coloring vs noise level. Right: example of a corrupted input and its reconstruction. The decoder consistently maps noisy input back toward valid COLORING graphs.

### 5.3 Quantitative performances of GRALE models on downstream tasks

**Graph classification/Regression.** We use the graph representations obtained by GRALE as input for classification and regression tasks in the MoleculeNet benchmark [70]. We compare our method to several graph representation learning baselines, including graph AutoEncoders (PIGVAE [68], VGAE [36], GraphMAE [**?** ]) and contrastive learning methods (Infograph [54], Simgrace [71]). For a fair comparison, all models are pre-trained on PUBCHEM 32, and the same predictive head is used for the downstream tasks. Detailed settings are provided in Appendix A. Overall, GRALE outperforms the other graph AutoEncoders and performs similarly, if not better, than the other baselines.

Table 3: Downstream tasks performance of different graph representation learning methods pretrained on PUBCHEM 32. We report the mean $\pm$ std over 5 train/test splits.

| MODEL | MLP REGRESSION (MAE ↓) | | SVR REGRESSION (MAE ↓) | | | SVC CLASSIF. (ROC-AUC ↑) | |
|---|---|---|---|---|---|---|---|
| | QM9 | QM40 | ESOL | LIPO | FREESOLV | BBBP | BACE |
| SIMGRACE | $0.110 \pm 0.003$ | $0.025 \pm 0.005$ | $0.293 \pm 0.020$ | $0.534 \pm 0.023$ | $0.374 \pm 0.008$ | $\underline{0.745} \pm 0.072$ | $\mathbf{0.866} \pm 0.050$ |
| INFOGRAPH | $0.122 \pm 0.001$ | $0.020 \pm 0.001$ | $\mathbf{0.255} \pm 0.016$ | $\mathbf{0.495} \pm 0.013$ | $0.297 \pm 0.010$ | $0.729 \pm 0.053$ | $0.845 \pm 0.046$ |
| GRAPHMAE | $0.222 \pm 0.016$ | $0.247 \pm 0.036$ | $0.291 \pm 0.012$ | $0.527 \pm 0.029$ | $0.378 \pm 0.028$ | $\mathbf{0.773} \pm 0.036$ | $\underline{0.857} \pm 0.039$ |
| GVAE | $0.765 \pm 0.005$ | $0.328 \pm 0.005$ | $0.306 \pm 0.027$ | $0.668 \pm 0.014$ | $0.344 \pm 0.022$ | $0.705 \pm 0.060$ | $0.771 \pm 0.049$ |
| PIGVAE | $\underline{0.031} \pm 0.001$ | $\underline{0.019} \pm 0.001$ | $0.279 \pm 0.020$ | $0.523 \pm 0.019$ | $\underline{0.283} \pm 0.013$ | $0.675 \pm 0.079$ | $0.816 \pm 0.049$ |
| GRALE | $\mathbf{0.015} \pm 0.001$ | $\mathbf{0.018} \pm 0.003$ | $\underline{0.274} \pm 0.014$ | $\underline{0.511} \pm 0.022$ | $\mathbf{0.272} \pm 0.017$ | $0.731 \pm 0.025$ | $0.821 \pm 0.051$ |

Table 4: Performances of different methods on graph prediction benchmarks from [37].

| MODEL | EDIT DIST. (↓) | | | | GI ACC. (↑) | | | |
|---|---|---|---|---|---|---|---|---|
| | COLORING 10 | COLORING 15 | QM9 | GDB13 | COLORING 10 | COLORING 15 | QM9 | GDB13 |
| FGWBARY | 6.73 | N.A. | 2.84 | N.A. | 01.00 | N.A. | 28.95 | N.A. |
| RELATIONFORMER | 5.47 | 2.64 | 3.80 | 7.45 | 18.14 | 21.99 | 09.95 | 00.05 |
| ANY2GRAPH | **0.19** | 1.22 | $\underline{2.61}$ | $\underline{3.63}$ | 84.44 | 43.77 | 29.85 | 16.25 |
| GRALE | 0.39 | $\underline{0.67}$ | 3.62 | 4.43 | $\underline{89.66}$ | $\underline{86.02}$ | $\underline{30.77}$ | $\underline{32.02}$ |
| GRALE + FINETUNING | $\underline{0.27}$ | **0.45** | **2.13** | **2.25** | **90.04** | **88.87** | **35.27** | **53.22** |

Table 5: We report (avg±std) edit distance and compute time over 1000 pairs of test graphs. All solvers use the default Pygmtools parameters except Greedy A$^*$ which is A$^*$ with beam width one.

| MODEL | EDIT DIST. (↓) | | COMPUTE TIME IN SECONDS (↓) | |
|---|---|---|---|---|
| | COLORING | PUBCHEM | COLORING | PUBCHEM |
| IPFP | $21.10 \pm 10.83$ | $40.66 \pm 12.16$ | $0.006 \pm 0.013$ | $\underline{0.073} \pm 0.052$ |
| RRWM | $20.66 \pm 10.90$ | $42.24 \pm 12.71$ | $0.540 \pm 0.190$ | $1.271 \pm 0.244$ |
| SM | $20.90 \pm 10.91$ | $43.20 \pm 12.86$ | $\mathbf{0.002} \pm 0.001$ | $0.076 \pm 0.022$ |
| GREEDY A$^*$ | $20.41 \pm 11.21$ | $\underline{32.53} \pm 08.15$ | $0.021 \pm 0.016$ | $0.110 \pm 0.054$ |
| A$^*$ | $19.66 \pm 11.01$ | N.A. | $3.487 \pm 1.928$ | $>100$ |
| GRALE | $\underline{08.92} \pm 05.61$ | $32.77 \pm 11.67$ | $\underline{0.005} \pm 0.001$ | $\mathbf{0.008} \pm 0.002$ |
| GRALE + FINETUNING | $\mathbf{07.64} \pm 04.42$ | $\mathbf{19.33} \pm 07.85$ | | |

**Graph prediction.** We now consider the challenging task of graph prediction, where the goal is to map an input $x \in \mathcal{X}$ to a target graph $G^*$. Following the surrogate regression strategy of [75, 8, 6], we train a surrogate predictor $\varphi : \mathcal{X} \mapsto \mathcal{Z}$ to minimize $\|\varphi(x) - g(G^*)\|_2^2$. At inference time, the predicted embedding is decoded into a graph using the pretrained decoder $\hat{G} = f \circ \varphi(x)$. We also consider a finetuning phase, where $\varphi$ and $f$ are jointly trained with the end-to-end loss $\mathcal{L}(f \circ \varphi(x), G^*)$. We evaluate this approach on the Fingerprint2Graph and Image2Graph tasks introduced in [37], using the same model for $\varphi$. Results are reported in Table 4. Thanks to pre-training (on COLORING for Image2Graph and PUBCHEM for Fingerprint2Graph), GRALE significantly outperforms prior methods on most tasks.

**Graph Matching.** Finally, we propose to use GRALE to match arbitrary graphs $G_1$ and $G_2$. To this end, we compute the node embeddings of both $G_1$ and $G_2$ and plug them into the matcher

$$T(G_1, G_2) = m\Big(g_{\text{nodes}}(G_1), g_{\text{nodes}}(G_2)\Big) \tag{11}$$

where we enforce that $T(G_1, G_2) \in \sigma_N$ by replacing the Sinkhorn algorithm with the Hungarian algorithm [15] inside the matcher. Then, we use this (potentially suboptimal) matching to compute an upper bound of the edit distance. In Table 5, we compare this approach to more classical methods as implemented in Pygmtools [62]. Pygmtools is fully GPU compatible, which enables a fair comparison with the proposed approach in terms of time complexity. The matcher operates out of distribution since it is trained to match input/output graphs that are expected to be similar $G_1 \approx G_2$ at convergence and here we sample random pairs of graphs. To mitigate this, we fine-tune the matcher on random pairs of graphs from the training set. As shown in Table 5, the proposed approach yields better upper bounds of the edit distance than classical solvers while being orders of magnitude faster.

## 6    Conclusion, limitations and future works

We introduced GRALE, a novel graph-level AutoEncoder, and showed that its embeddings capture intrinsic properties of the input graphs, achieving SOTA performance and beyond, across numerous

tasks. Trained on large-scale molecule data, GRALE provides a strong foundation for solving graph-level problems through vectorial embeddings. Arguably, GRALE's main limitation is its computational complexity (mostly cubic), which led us to restrict graphs to $N = 32$. This could be mitigated in future work using approximate attention in the Evoformer-based modules [2, 11, 13] and accelerated Sinkhorn differentiation in the matcher [5, 16]. Beyond this, GRALE opens up several unexplored directions. For instance, GRALE could serve as the basis for a new graph variational autoencoder for generative tasks. Finally, given its strong performance in graph prediction, applying GRALE to the flagship task of molecular elucidation [34, 43] appears to be a promising next step.

## Acknowledgments and Disclosure of Funding

This project was provided with computing AI and storage resources by GENCI at IDRIS thanks to the grant 2025-AD011016098 on the supercomputer Jean Zay's H100 partition. It received funding from the European Union's Horizon Europe research and innovation programme under grant agreement 101120237 (ELIAS). Views and opinions expressed are however those of the authors only and do not necessarily reflect those of the European Union or European Commission. Neither the European Union nor the granting authority can be held responsible for them. This research was also supported in part by the French National Research Agency (ANR) through the PEPR IA FOUNDRY project (ANR-23-PEIA-0003), the MATTER project (ANR-23-ERCC-0006-01) and ANR/France 2030 program (ANR-23-IACL-0005). It also received funding from the Hi! PARIS and Fondation de l'École polytechnique. Finally, the authors would like to thank the Isaac Newton Institute for Mathematical Sciences, Cambridge, for support and hospitality during the programme "Representing, calibrating & leveraging prediction uncertainty from statistics to machine learning" where work on this paper was undertaken. This work was supported by EPSRC grant no EP/Z000580/1.

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

# A   Experimental setting

## A.1   Datasets

**COLORING.**   COLORING is a suite of synthetic datasets introduced in [37] for benchmarking graph prediction methods. Each sample consists of a pair *(Image, Graph)* as illustrated in Figure 7. Importantly, all COLORING graphs satisfy the 4-color theorem, that is, no adjacent nodes share the same color (label). Since it is a synthetic dataset, one can generate as many samples as needed to create the train/test set. We refer to the original paper for more details on the dataset generation. Note that, unless specified otherwise, we ignore the images and consider COLORING as a pure graph dataset.

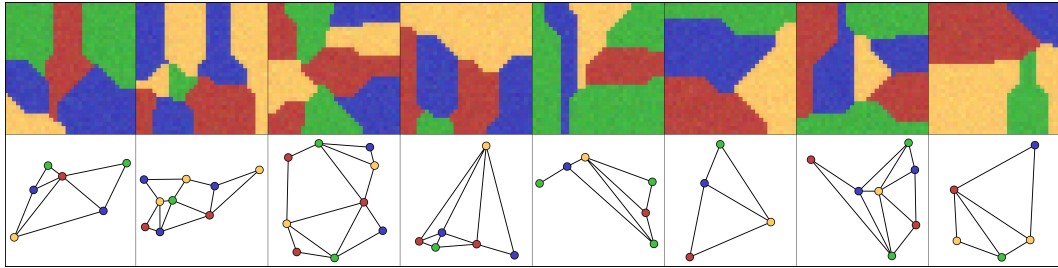

Figure 7: Image/Graph pairs from the COLORING dataset (courtesy of [37]).

**Molecular Datasets.**   To go beyond synthetic data, we also consider molecular datasets. They represent a very interesting application of GRALE as 1) the graphs tend to be relatively small, 2) a very large amount of unsupervised data is available for training, and 3) the learned representation can be applied to many challenging downstream tasks. Most molecular datasets are stored as a list of SMILES strings. In all cases, we use the same preprocessing pipeline. First, we convert the SMILES string to graphs using Rdkit, and we discard the datapoint if the conversion fails. Then we remove the hydrogen atoms and use the remaining atom numbers as node labels and bond type (none, single, double, triple, or aromatic) as edge labels. Finally, we remove all graphs with more than $N = 32$ atoms to save computations.

**Training Datasets.**   We train GRALE on 3 different datasets. First of all, we train on COLORING 20, a variant of COLORING where we sample 300k graphs of size ranging from 5 to 20. This version of GRALE is applied to all downstream tasks related to COLORING. Then, we also train on PUBCHEM 32, a large molecular dataset that we obtained by downloading all molecules, with up to 32 heavy atoms from the PUBCHEM database [33]. This version is applied to all downstream tasks related to molecules. For the ablation studies, we reduce computation by training a smaller version of the model on PUBCHEM 16, a truncated version of PUBCHEM 32 that retains only graphs with up to 16 nodes. Table 6 presents the main statistics for each dataset.

Table 6: Training Datasets.

| DATASET | GRAPH SIZE: AVG± STD | GRAPH SIZE: MAX | N SAMPLES |
|---|---|---|---|
| COLORING 20 | $12.50 \pm 4.33$ | 20 | 300k |
| PUBCHEM 16 | $13.82 \pm 2.17$ | 16 | 14M |
| PUBCHEM 32 | $22.62 \pm 5.79$ | 32 | 84M |

**Downstream Datasets: Classification/Regression.**   For classification and regression downstream tasks, we consider supervised molecular datasets from the MoleculeNet benchmark [70]. We choose datasets that cover a wide range of fields from Quantum Mechanics (QM9, QM40 [4]), to Physical Chemistry (Esol, Lipo, Freesolv), to Biophysics (BACE), to Physiology (BBBP). In all cases, we preprocess the molecules using the same pipeline as for PUBCHEM 32 to enable transfer learning. In

---

[4]Out of the many available regression targets we focus only on internal energy.

particular, we discard all graphs with more than $N = 32$ atoms, resulting in a truncated version of the datasets. This motivates us to sample new random train/test splits (90%/10%). For regression tasks, we also normalize the target with a mean of 0 and a variance of 1. We provide the main statistics of those datasets in Table 7.

Table 7: Downstream datasets (Classification/Regression). We also report the number of samples in the original dataset, before the truncation due to the preprocessing pipeline.

| DATASET | GRAPH SIZE: AVG± STD | GRAPH SIZE: MAX | N SAMPLES | N SAMPLES ORIGINAL |
|---------|---------------------|-----------------|-----------|--------------------|
| QM9 | $8.80 \pm 0.51$ | 9 | 133885 | 133885 |
| QM40 | $21.06 \pm 6.26$ | 32 | 137381 | 162954 |
| ESOL | $13.06 \pm 6.40$ | 32 | 1118 | 1118 |
| LIPO | $24.12 \pm 5.53$ | 32 | 3187 | 4200 |
| FREESOLV | $8.72 \pm 4.19$ | 24 | 642 | 642 |
| BBBP | $21.10 \pm 6.38$ | 32 | 1686 | 2050 |
| BACE | $27.23 \pm 3.80$ | 32 | 735 | 1513 |

**Downstream Datasets: Graph Prediction.**   For graph prediction, we select two challenging tasks proposed in [37]. First of all, we consider the *Image2Graph* task where the goal is to map the image representation of a COLORING instance to its graph representation, meaning that the inputs are the first row of Figure 7 and the targets are the second row. For this task, we use COLORING 10 and COLORING 15, which are referred to as COLORING medium and COLORING big in the original paper. Secondly, we consider a Fingerprint2Graph task. Here, the goal is to reconstruct a molecule from its fingerprint representation [57], that is, a list of substructures. Once again, we consider the same molecular datasets as proposed in the original article, namely QM9 and GDB13 [4]. Table 8 presents the main statistics of those datasets.

Table 8: Downstream datasets (Graph prediction).

| DATASET | GRAPH SIZE: AVG± STD | GRAPH SIZE: MAX | N SAMPLES |
|---------|---------------------|-----------------|-----------|
| COLORING 10 | $7.52 \pm 1.71$ | 10 | 100K |
| COLORING 15 | $9.96 \pm 3.15$ | 15 | 200K |
| QM9 | $8.79 \pm 0.51$ | 9 | 120K |
| GDB13 | $12.76 \pm 0.55$ | 13 | 1.3M |

## A.2   Model and training hyperparameters

As detailed in the previous section, we train 3 variants of our models, respectively, on COLORING 20, PUBCHEM 32 and PUBCHEM 16. We report the hyperparameters used in Table 9. Note that to reduce the number of hyperparameters, we set the number of layers in all modules (evoformer encoder, transformer decoder, transformer encoder, evoformer decoder) to the same value $L$, idem for the number of attention heads $H$ and node/edge hidden dimensions. In all cases, we trained with ADAM [35], a warm-up phase, and cosine annealing. We also report the number of GPUs and the total time required to train the models with these parameters.

## A.3   Graph reconstruction metrics

To measure the error between a predicted graph $\hat{G}$ and a target graph $G$ we first consider the graph edit distance [22, 51], that is the number of modification (editions) that should be applied to get to $\hat{G}$ from $G$ (and vice-versa). The possible editions are node or edge addition, deletion, or modification, where node/edge modification stands for changing the label of a node/edge. In this paper, we set the cost of all editions to 1. It is well known that computing the edit distance is equivalent to solving a graph matching problem [1]. For instance, in the case of two graphs of the same size $G = (F, C)$ and $\hat{G} = (\hat{F}, \hat{C})$, the graph edit distance is written as

Table 9: For every dataset used to train GRALE, we report: 1) The architecture parameters, 2) The training Parameters, 3) The computational resources required. Note that when the hidden dimension of MLPs is set to "None", the MLPs are replaced by a linear layer plus a ReLU activation. This makes the model much more lightweight.

| TRAINING DATASET | COLORING | PUBCHEM 16 | PUBCHEM 32 |
|---|---|---|---|
| MAXIMUM OUTPUT SIZE N | 20 | 16 | 32 |
| NUMBER OF TOKENS $K$ | 4 | 8 | 16 |
| DIMENSION OF TOKENS $D$ | 32 | 32 | 32 |
| TOTAL EMBEDDING DIM $d = K \times D$ | 128 | 256 | 512 |
| NUMBER OF LAYERS | 5 | 5 | 7 |
| NUMBER OF ATTENTION HEADS | 4 | 4 | 8 |
| NODE DIMENSIONS | 128 | 128 | 256 |
| NODE HIDDEN DIMENSIONS (MLPS) | NONE | 128 | 256 |
| EDGE DIMENSIONS | 64 | 64 | 128 |
| EDGE HIDDEN DIMENSIONS (MLPS) | NONE | NONE | NONE |
| TOTAL PARAMETER COUNT | 2.0M | 2.5M | 11.7M |
| NUMBER OF GRADIENT STEPS | 300K | 700K | 1.5 M |
| BATCH SIZE | 64 | 128 | 256 |
| EPOCHS | 64 | 5 | 5 |
| NUMBER OF WARMUP STEPS | 4K | 8K | 16K |
| BASE LEARNING RATE | 0.0001 | 0.0001 | 0.0001 |
| GRADIENT NORM CLIPPING | 0.1 | 0.1 | 0.1 |
| NUMBER OF GPUS (L40S) | 1 | 1 | 2 |
| TRAINING TIME | 8H | 20H | 100H |

$$\texttt{Edit}(G, \hat{G}) = \min_{P \in \sigma_N} \mathcal{L}_{\text{ALIGN}}(G, P[\hat{G}]), \tag{12}$$

where $P[\hat{G}] = (P\hat{F}, P\hat{C}P^T)$ and

$$\mathcal{L}_{\text{ALIGN}}(G, \hat{G}) = \sum_i \mathbf{1}[F_i \neq \hat{F}_i] + \sum_{i,j} \mathbf{1}[C_{i,j} \neq \hat{C}_{i,j}] \tag{13}$$

Note that this rewrites as a Quadratic Assignment Problem (QAP) known as Lawler's QAP

$$\texttt{Edit}(G, \hat{G}) = \min_{P \in \sigma_N} \text{vect}(P)^T K \text{vect}(P) \tag{14}$$

For the proper choice of $K \in \mathbb{R}^{N^2 \times N^2}$. This formulation can be extended to cases where $G$ and $\hat{G}$ have different sizes, up to the proper padding of $K$, which is equivalent to padding the graph directly as we do in this paper [24]. Since the average edit distance that we observe with GRALE is typically lower than 1, we also report a more challenging metric, the Graph Isomorphism Accuracy (GI Acc.), also known as top-1-error or hit1, that measures the percentage of samples with edit distance 0.

$$\texttt{Acc}(G, \hat{G}) = \mathbf{1}[\texttt{Edit}(G, \hat{G}) = 0] \tag{15}$$

# B   Additionnal experiments

## B.1   Training Complexity

When it comes to molecules, unsupervised data is massively available. At the time this paper was written, the PubChem database [33] contained more than 120 million compounds, and this number is increasing. In the context of this paper, this raises the question of how much of this data is necessary

to train the model. To answer this, we propose to train GRALE on a truncated dataset and observe the performance. Once again, we train on PUBCHEM 16 using the medium-sized model described in Table 9 to reduce computational cost. Note that we keep the size of the model fixed. For more in-depth results on neural scaling laws in molecular representation learning, we refer to [10].

We propose 2 different performance measures. On the one hand, we report the quality of reconstruction (on a test set) against the size of the data set (Figure 8, left). On the other hand, we report the results achieved when the learned representation is applied to a downstream task. More precisely, we report the MAE observed when the learned representation is used for regression on the QM9 dataset (Figure 8, right).

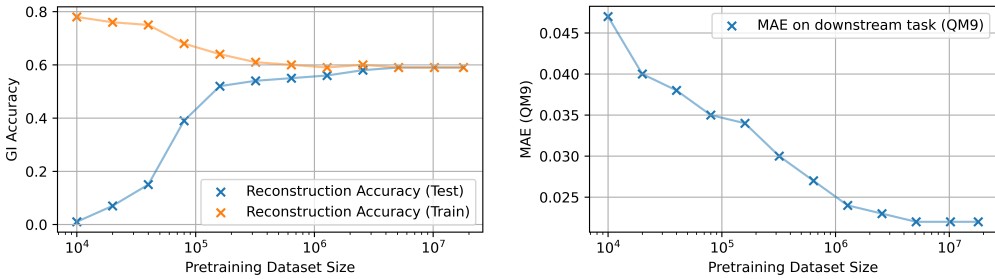

Figure 8: GRALE performances vs pretraining dataset size (in log scale). Left: Train/test reconstruction accuracy. Right: Downstream performance on the QM9 regression task using the learned embeddings.

For the reconstruction accuracy, we observe overfitting for all datasets smaller than one million molecules. More interestingly, we observe that the performance on the downstream task is also highly dependent on the size of the pretraining dataset, as the performances keep improving past one million samples. Overall, it appears that this version of GRALE is able to leverage a very large molecular dataset. To apply GRALE to a different field, where pretraining datasets are smaller, it might require considering a different set of parameters of perhaps a more frugal architecture. For instance, the encoder and decoder could be replaced by the more lightweight baselines proposed in E.

## B.2 Learning to AutoEncode as a pretraining task and the choice of the latent dimension

The original motivation for the AutoEncoder is dimensionality reduction: given an input $x \in \mathbb{R}^{d_x}$, the goal is to find some equivalent representation $z \in \mathbb{R}^d$ such that $d \ll d_x$. More generally, learning to encode/decode data in a small latent space is a challenging unsupervised task, well-suited for pretraining. However, in the case of graphs, the original data $x$ is not a vector, which makes it hard to estimate what is a "small" latent dimension $d$. To explore this question, we propose to train GRALE for various values of $d$ and report the corresponding performances. As in the previous experiment, we report both the reconstruction accuracy and the downstream performance on QM9 (Figure 9).

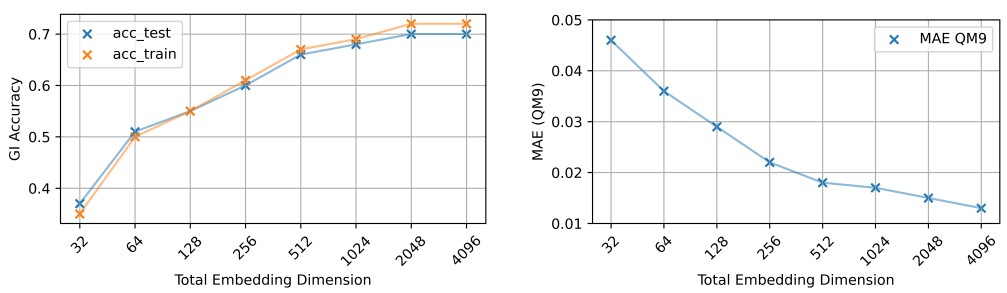

Figure 9: GRALE performances vs total embedding dimension $d$ (in log scale). Left: Train/test reconstruction accuracy. Right: Downstream performance on the QM9 regression task using the learned embeddings.

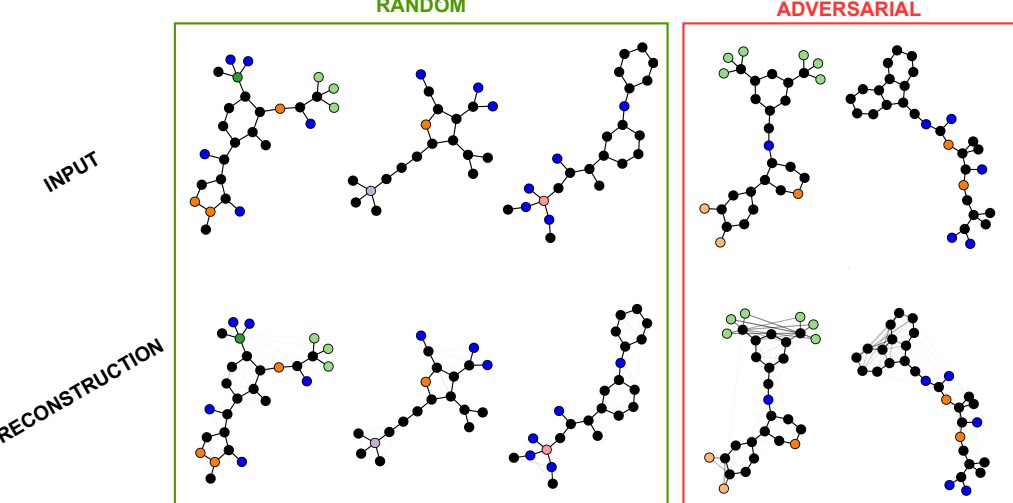

Figure 10: Inputs from the PUBCHEM 32 dataset along with their GRALE reconstruction. Node color represents the atomic number: Black for carbon, Blue for Oxygen, etc. The different types of bonds are not represented, but we represent the predicted probability for edge existence with its width. On the left we provide random samples for reference. On the right we provide two "failure case" (sampled in an adversarial fashion from PUBCHEM 32). In both adversarial molecule, the input graph exhibits many symmetries. For instance, in the second sample, 6 nodes are perfectly symmetric (in the sense of the Weisfeiler-Leman test).

First, we observe that reconstruction performance and downstream task performance are highly correlated, which is consistent with the observation made in the previous experiment and the results of this work in general. This confirms the intuition that learning to encode and decode a graph is a suitable pretraining task. We also observe that small embedding dimensions act as a regularizer, preventing overfitting in terms of graph reconstruction accuracy. Despite this, we do not observe that the downstream performance deteriorates with higher embedding dimensions[5]. This suggests that learning to encode/decode entire graphs into an Euclidean space is always a challenging task, even when the latent space is of large dimensions.

### B.3 Reconstruction failure case

We now highlight an interesting failure case of the AutoEncoder. To this end, we plot the pair of inputs and outputs with the maximum reconstruction error. We observe that the hardest cases to handle for the AutoEncoder are those where the input graph exhibits many symmetries. When this happens, it becomes difficult for the matcher to predict the optimal matching between input and output. The main reason for this is that similar nodes have similar embeddings, and since the matcher is based on these hidden node features, it might easily confuse nodes with apparently similar positions in the graph. As a result, GRALE has difficulty converging in this region of the graph space. To illustrate this phenomenon, we select the two graphs with the highest reconstruction error out of 1000 random test samples and plot their reconstruction (Figure 10). For comparison, we also provide 3 random samples that are correctly reconstructed.

### B.4 Latent space exploration

Finally, we conclude this section with a qualitative exploration of the latent space learned by GRALE. In Figure 11, we first plot the two principal PCA components of the latent space learned for COL-ORING 20 and PUBCHEM 32. Remarkably, this reveals that the first PCA component explicitly encodes the size of the graph (number of nodes). This observation motivates us to investigate the interpretation of the second PCA direction. To this end, we sample 10,000 graphs from each dataset and select the four graphs whose embeddings maximize and minimize the projection along the second

---

[5]Within this "reasonable" range of values.

PCA component. We visualize these graphs in Figure 12. In both cases, we observe that the second component appears to encode properties related to node labels, such as the number of carbon atoms for PUBCHEM 32. This aligns with the observations of Krzakala et al. [37], who report that graph prediction models tend to encode node labels very strongly.

It is unclear whether this behavior is desirable in practice. For instance, in the case of molecular data, it might be preferable for the latent space to be structured according to chemical properties instead. To further explore this aspect, we focus on the QM9 dataset. QM9 mostly contains molecules of size 9, which allows us to zoom in on a subspace corresponding to molecules of a fixed size. Moreover, this dataset provides a wide range of ground-truth chemical properties. We randomly sample 10,000 molecules with exactly 9 nodes from QM9, encode them using a version of GRALE pretrained on PUBCHEM 32, and apply t-SNE to reduce the embeddings to two dimensions. In Figure 13, we color the resulting 2D latent space according to various chemical properties. Overall, we observe that these properties exhibit some degree of structure in the latent space, although this structure is less pronounced than that associated with the structural properties discussed above.

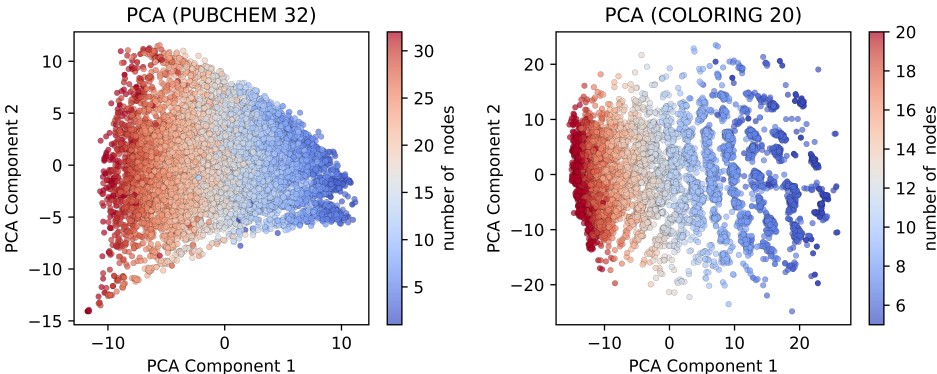

Figure 11: PCA in the latent space for PUBCHEM 32 (left) and COLORING (right). Each point corresponds to the embedding of one graph. Points are colored according to the number of nodes in the graph. In both cases, the first principal component appears to encode the graph size.

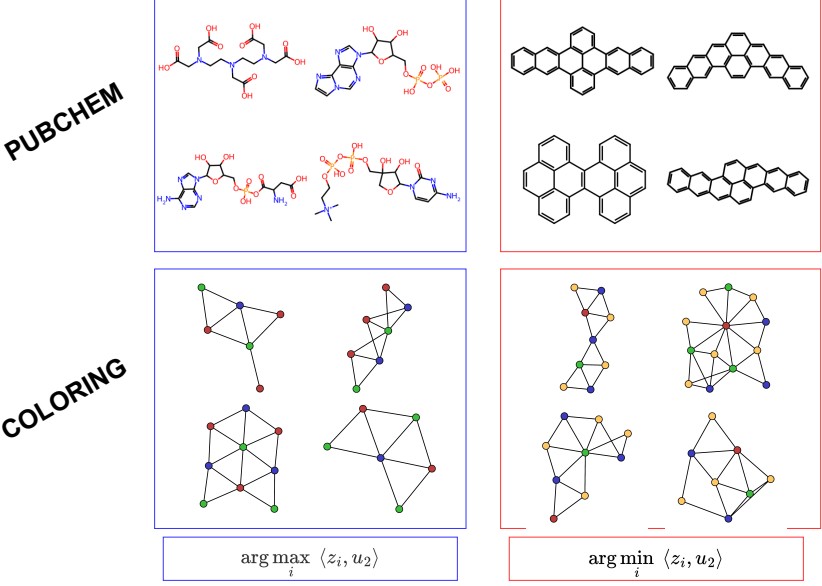

Figure 12: To reveal the role of the second principal component ($u_2$), we retrieve the graphs whose embeddings maximize and minimize the projection along this direction. For PUBCHEM (resp. COLORING), this component appears to roughly encode the number of carbon atoms (resp. yellow nodes).

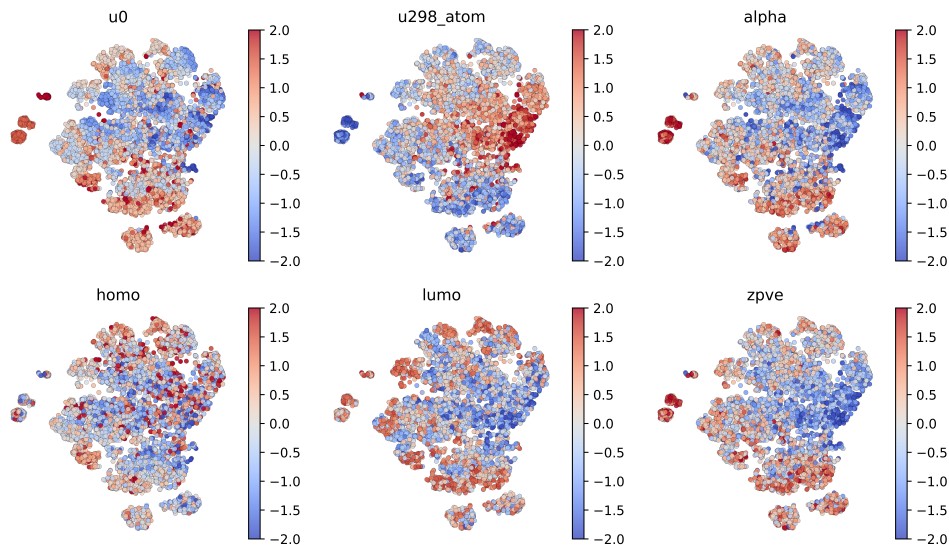

Figure 13: Two-dimensional t-SNE of the QM9 embedding space, with perplexity set to 50. Each point corresponds to the embedding of one graph, colored according to different chemical properties.

## C  GRALE architecture details

### C.1  Featurizer

There are many ways to parametrize the featurizers $\phi$ and $\phi'$. In this paper, we focus on simple choices that have already demonstrated good empirical performances [45, 77, 68, 37]. From a datapoint $x \in \mathcal{D}$, the featurizer first constructs a node label matrix $F_0(x) \in \mathbb{R}^{n \times d_0}$, an adjacency matrix $A(x) \in \{0,1\}^{n \times n}$ and a shortest path matrix $SP(x) \in \mathbb{N}^{n \times n}$ where $n$ is the size of the graph. We then augment the node features with k-th order diffusion

$$F(x) = \text{CONCAT}[F_0(x), A(x)F_0(x), ..., A^k(x)F_0(x)] \tag{16}$$

where $k$ is an hyperparameter set to $k = 2$ in our experiments. Then the edge features are defined as

$$C_{i,j}(x) = \text{CONCAT}[F_i(x), F_j(x), \text{ONE-HOT}(A_{i,j}(x)), \text{PE}(SP_{i,j}(x))] \tag{17}$$

where ONE-HOT denotes one-hot encoding and PE denotes sinusoidal positionnal encoding [58]. If edge labels are available, they are concatenated to $C$ as well. Finally, the featurizers are defined as

$$\phi(x) = (F(x) + \text{Noise}, C(x)), \quad \phi'(x) = \text{PADDING}(F(x), C(x)) \tag{18}$$

where Noise is a random noise matrix, and PADDING pads all graphs to the same size $N > n$ as defined above. The noise component breaks the symmetries in the input graph, which enables the encoder to produce distinct embeddings for all the nodes. We demonstrate empirically that this is crucial for the performance of the matcher and provide a more qualitative explanation in B.3. We leave the exploration of more complex, and possibly more asymmetric, featurizers to future work.

### C.2  Encoder

The encoder $g$ takes as input a graph $G = (F, C)$ and returns both the node level embeddings $g_{\text{nodes}}(G) = X$ and graph level embedding $g_{\text{graph}}(G) = Z$. The main component of $g$ is a stack of $L$ Evoformer Encoder layers [30] that produces the hidden representation $(F^L, C^L)$ of the input graph.

$$(F^{l+1}, C^{l+1}) = \text{EvoformerEncoder}(F^l, C^l) \tag{19}$$

where $F^1 \in \mathbb{R}^{n \times d_F}$ (resp. $C^1 \in \mathbb{R}^{n \times n \times d_c}$) are initialized by applying a node-wise (resp. edge-wise) linear layer on $F$ (resp. $C$). The Evoformer Encoder layer used in GRALE is represented in Figure 14. Compared to the original implementation, we make two notable changes that make this version much more lightweight. First, we replace the MLP of the Feed Forward Block (FFB) with a simple linear layer followed by an activation function. Then, of the 4 modules dedicated to the update of $C$, we keep only one that we call Triangular Self Attention Block (tSAB) to highlight the symmetry with a Transformer Encoder. The definitions for all these blocks are provided in Appendix D.

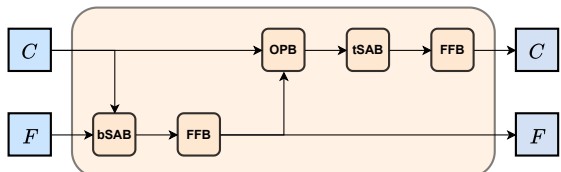

Figure 14: Architecture of the Evoformer Encoder layer used for encoding the input graph $G = (F, C)$.

Once the hidden representation $(F^L, C^L)$ has been computed, the node level embeddings are simply derived from a linear operation

$$X_i = \texttt{Linear}(F_i^L) \quad \text{if} \quad i < n, \quad X_i = u \quad \text{otherwise.} \tag{20}$$

where $u$ is a learnable padding vector.

Finally, the graph-level representation is obtained by pooling $C$ with a Transformer Decoder. More precisely, we first flatten $C$ to be a $n^2 \times d_C$ matrix, then we pass it to a standard $L$ layer Transformer Decoder to output the graph level embedding $Z = Z_Q^L \in \mathbb{R}^{K \times D}$.

$$Z_Q^{l+1} = \texttt{TransformerDecoder}(Z_Q^l, C^L) \tag{21}$$

where $Z_Q^0$ is a learnable query matrix in $\mathbb{R}^{K \times D}$. This module is illustrated in Figure 15.

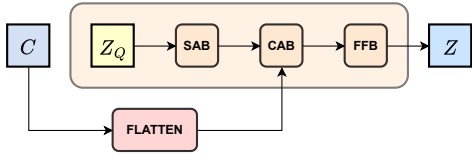

Figure 15: Architecture of the Transformer Decoder used for pooling $C^L$.

### C.3 Decoder

The decoder takes as input the graph embedding $Z$ and should output both the reconstruction $\hat{G} = (\hat{h}, \hat{F}, \hat{C})$ and the node embeddings $\hat{X}$ that are required to match input and output graphs. To this end, the proposed architecture mimics that of a Transformer Encoder-Decoder except that we replace the usual Transformer Decoder with a novel Evoformer Decoder. More formally, the latent representation $Z = Z^0$ is first updated by a Transformer Encoder with $L$ layers

$$Z^{l+1} = \texttt{TransformerEncoder}(Z^l). \tag{22}$$

For completeness, we also recall the content of a transformer encoder layer in Figure 16.

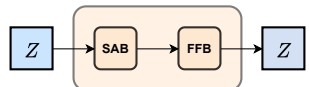

Figure 16: Architecture of the Transformer Encoder.

Then, similarly to a Transformer Decoder, we define a learnable graph query $(F_Q^0, C_Q^0)$ where $F_Q^0 \in \mathbb{R}^{N \times d_F}$ and $C_Q^0 \in \mathbb{R}^{N \times N \times d_C}$ that we update using L layers of the novel Evoformer Decoder

$$(F_Q^{l+1}, C_Q^{l+1}) = \texttt{EvoformerDecoder}(F_Q^l, C_Q^l, Z^L) \tag{23}$$

The proposed Evoformer Decoder is to the Evoformer Encoder, the same as the Transformer Decoder is to the Transformer Encoder, i.e., it is the same, plus up to a few Cross-Attention Blocks (CAB) that enable conditioning on $Z^L$. See Figure 17 for the details of the proposed layer.

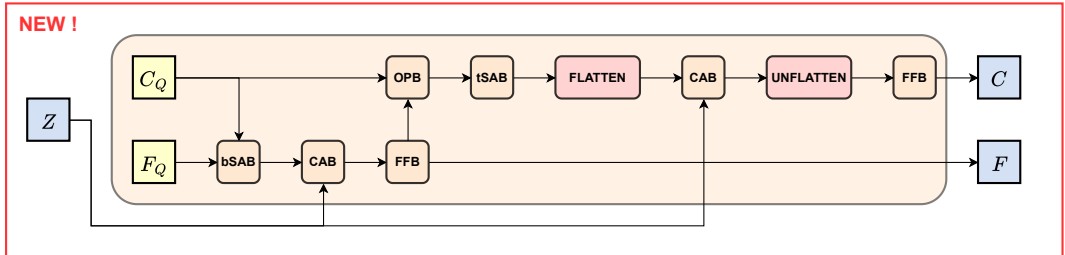

Figure 17: Similarly to a Transformer Decoder, the proposed Evoformer Decoder layer augments the Evoformer Encoder with 2 Cross-Attention Blocks (CAB) so that the output can be conditioned on some source $Z$. All the inner blocks are defined in D.

Finally, the reconstructed graphs are obtained with a few linear heads

$$\hat{h} = \texttt{Sigmoid}(\texttt{Linear}(F_Q^L)), \quad \hat{F} = \texttt{Linear}(F_Q^L), \quad \hat{C} = \texttt{Linear}(C_Q^L) \tag{24}$$

where the linear layers are applied node and edge-wise. Similarly, the node-level embeddings of the output graphs are defined as

$$\hat{X} = \texttt{Linear}(F_Q^L) \tag{25}$$

### C.4   Matcher

The matcher uses the node embeddings of the input graph $X$ and target graph $\hat{X}$ to compute the matching $\hat{T}$ between the two graphs. The first step is to build an affinity matrix $K \in \mathbb{R}^{N \times N}$ between the nodes. We propose to parametrize $K$ using two one hidden layer MLPs $\texttt{MLP}_{in}$ and $\texttt{MLP}_{out}$

$$K_{i,j} = \exp(-|\texttt{MLP}_{in}(X_i) - \texttt{MLP}_{out}(\hat{X}_j)|) \tag{26}$$

Note that $K$ is positive, but might not be a bistochastic matrix. To this end, we project $K$ on $\sigma_N$ using Sinkhorn projections

$$\hat{T} = \texttt{SINKHORN}(K) \tag{27}$$

We fix the number of Sinkhorn steps to 100, and to ensure stability, we perform the iterations in the log domain [12]. At train time, we backpropagate through Sinkhorn by unrolling through these iterations [16]. At test time, we fully replace Sinkhorn by the Hungarian algorithm [15] to ensure that the matching $\hat{T}$ is a discrete, permutation matrix.

### C.5   Loss

Recall that we use the loss originally introduced in Any2Graph [37]:

$$\mathcal{L}_{\text{OT}}(G, \hat{G}, T) = \sum_{i,j}^N \ell_h(h_i, \hat{h}_j) T_{i,j} + \sum_{i,j}^N h_i \ell_F(F_i, \hat{F}_j) T_{i,j} + \sum_{i,j,k,l}^N h_i h_k \ell_C(C_{i,k}, \hat{C}_{j,l}) T_{i,j} T_{k,l}, \tag{28}$$

where the ground losses $\ell_h, \ell_F$ and $\ell_C$ still need to be defined. We decompose the node reconstruction loss $\ell_F$ between the part devoted to node labels (discrete) $\ell_F^d$ and node features (continuous) $\ell_F^c$. Similarly, we decompose $\ell_C$ into $\ell_C^d$ and $\ell_C^c$. Following Any2Graph [37], we use cross-entropy loss for all discrete losses $\ell_h, \ell_F^d, \ell_C^d$ and L2 loss for all continuous losses $\ell_F^c, \ell_C^c$. This makes a total of 5 terms that we balance using 5 hyperparameters $\alpha_h, \alpha_F^d, \alpha_F^c, \alpha_C^d$ and $\alpha_C^c$. Once again, we follow the guidelines derived in the original paper and set

$$\begin{cases} \alpha_h = \frac{1}{N}, \\ \alpha_F^d = \frac{1}{N}, \\ \alpha_F^c = \frac{1}{2N}, \\ \alpha_C^d = \frac{1}{N^2}, \\ \alpha_C^c = \frac{1}{2N^2}, \end{cases} \tag{29}$$

# D   Definitions of attention based models inner blocks

For completeness, we devote this section to the definition of all blocks that appear in the modules used in GRALE. For more details, we refer to specialized works such as Lee et al. [40] for Transformer and Jumper et al. [30] for Evoformer.

**From layers to blocks.**   We adopt the convention that a block is always made of a layer plus a normalization and a skip connection.

$$\texttt{BLOCK}(x) = \texttt{NORM}(x + \texttt{LAYER}(x)) \tag{30}$$

Note that, in some works, the layer normalization is placed at the start of the block instead [72].

**Parallelization.**   To lighten the notations we denote $f[X]$ whenever function $f$ is applied to the last dimension of $X$. For instance, when $X \in \mathbb{R}^{N \times D}$ is a feature matrix ($N$ nodes with features of dimension $D$), we have $f[X]_i = f(X_i)$. Similarly, when $C \in \mathbb{R}^{N \times N \times D}$ is an edge feature tensor, we have $f[C]_{i,j} = f(C_{i,j})$.

**Subscripts convention.**   In the following, we use the subscripts $i, j, k$ for the nodes/tokens indexing, $a, b, c$ for features dimensions and $l$ for indexing the heads in multi-head attention.

## D.1   Node level blocks

**Feed-Forward Block** (FFB).   Given an input $X \in \mathbb{R}^{N \times D}$, the FF layer simply consist in applying the same MLP to all lines/tokens/nodes of $X$ in parallel

$$\texttt{FFB}(X) = \texttt{NORM}(X + \texttt{MLP}[X]) \tag{31}$$

By construction, the FFB block is permutation equivariant w.r.t. $X$.

**Dot Product Attention.**   Given a query matrix $Q \in \mathbb{R}^{N \times D}$, key and value matrices $K, V \in \mathbb{R}^{M \times D}$ and biais matrix $B \in \mathbb{R}^{N \times M}$, the Dot Product Attention attention writes as:

$$\texttt{DPA}(Q, K, V, B) = \texttt{Softmax}[QK^T + B]V \tag{32}$$

More generally, for $B \in \mathbb{R}^{N \times M \times h}$, Multi-Head Attention writes as

$$\texttt{MHA}(Q, K, V, B) = \texttt{CONCAT}(O_1, ..., O_h) \tag{33}$$

where

$$O_l = \texttt{DPA}\big(q_l[Q], k_l[K], v_l[V], b^l\big) \tag{34}$$

and are linear layers and $b_{i,j}^l = B_{i,j,l}$.

**Cross-Attention Block (`CAB`).** Given $X \in \mathbb{R}^{N \times D}$ and $Y \in \mathbb{R}^{M \times D}$ we define:

$$\texttt{CA}(X, Y) = \texttt{MHA}\big(q[X], k[Y], v[Y], 0\big) \tag{35}$$

where $q, k, v : \mathbb{R}^D \mapsto \mathbb{R}^D$ are linear layers. The Cross-Attention Block is:

$$\texttt{CAB}(X, Y) = \texttt{NORM}(X + \texttt{CA}(X, Y)) \tag{36}$$

The Cross Attention layer is permutation equivariant with respect to $X$ and invariant with respect to the permutation of context $Y$.

**Self-Attention Block (`SAB`).** Given $X \in \mathbb{R}^{N \times D}$, the Self-Attention Block writes as

$$\texttt{SAB}(X) = \texttt{CAB}(X, X) \tag{37}$$

The Self Attention layer is permutation equivariant with respect to $X$.

### D.2 Graph level blocks

**Einstein Notations.** In the following, we adopt the Einstein summation convention for tensor operations. For instance, given matrices $A \in \mathbb{R}^{N \times D}$ and $B \in \mathbb{R}^{D \times M}$, the matrix multiplication $C = AB$, defined as $C_{i,j} = \sum_k A_{i,k} B_{k,j}$, is denoted compactly as $C_{i,j} = A_{i,k} B_{k,j}$ and the unused indices are implicitly summed over.

**Triangular Attention.** Triangle Attention is the equivalent of self-attention for the edges. To reduce the size of the attention matrix, edge $(i, j)$ can only attend to its neighbouring edges $(i, k)$. Thus for $Q, K, V \in \mathbb{R}^{N \times N \times D}$, the triangle attention layer writes as:

$$\texttt{TA}(Q, K, V)_{i,j,a} = A_{i,j,k} V_{i,k,a} \tag{38}$$

where the $A_{i,j,k}$ is the attention between $(i, j)$ and $(i, k)$ defined as $A_{i,j,k} = \texttt{Softmax}(Q_{i,j,a} K_{i,k,a})$. Multi-head attention can be defined in the same way as for the self-attention layer. Note that the original Evoformer also includes 3 similar layers where $(i, j)$ can only attend to $(k, i), (k, j)$ and $(j, k)$. For the sake of simplicity, we remove those layers in our implementation.

**triangular Self-Attention Block (`tSAB`).** Denoting $C \in \mathbb{R}^{N \times N \times D}$, we define:

$$\texttt{tSA}(C) = \texttt{TA}\big(q[C], k[C], v[C]\big) \tag{39}$$

where $q, k, v : \mathbb{R}^D \mapsto \mathbb{R}^D$ are linear layers. The triangular Self-Attention Block is defined as:

$$\texttt{TSAB}(C) = \texttt{NORM}(C + \texttt{tSA}(C)) \tag{40}$$

The triangular self-attention layer satisfies second-order permutation equivariance with respect to $C$.

**Outer Product Block (`OPB`).** Given a node feature matrix $X \in \mathbb{R}^{N \times D}$ and edge feature tensor $C \in \mathbb{R}^{N \times N \times D}$, the Outer Product layer enables information flow from the nodes to the edges.

$$\texttt{OP}(X)_{i,j,c} = X_{i,a} W_{a,b,c} X_{j,b} \tag{41}$$

where $W \in \mathbb{R}^{D \times D \times D}$ is a learnable weight tensor. The Outer Product Block is defined as:

$$\texttt{OPB}(C, X) = \texttt{NORM}(C + \texttt{OP}(X)) \tag{42}$$

**biased Self-Attention Block** (bSAB). Conversely, the biased self-attention layer enables information flow from the edges to the nodes. Given a node feature matrix $X \in \mathbb{R}^{N \times D}$ and edge feature tensor $C \in \mathbb{R}^{N \times N \times D}$ we define:

$$\text{bSA}(X, C) = \text{MHA}\big(q[X], k[X], v[X], b[C]\big) \tag{43}$$

where $q, k, v : \mathbb{R}^D \mapsto \mathbb{R}^D$ and $b : \mathbb{R}^D \mapsto \mathbb{R}^h$ are linear layers. Finally, the biased Self-Attention Block is:

$$\text{bSAB}(X, C) = \text{NORM}(X + \text{bSA}(X, C)) \tag{44}$$

## E  Ablation studies details

In section 5.1, we conduct an extensive ablation study where we validate the choice of our model components by replacing them with a baseline. We now provide more precise details on the baselines used for this experiment.

**Loss.** Recall the expression of the loss we propose for GRALE:

$$\mathcal{L}_{\text{OT}}(G, \hat{G}, \hat{T}) = \sum_{i,j}^{N} \ell_h(h_i, \hat{h}_j)\hat{T}_{i,j} + \sum_{i,j}^{N} h_i \ell_F(F_i, \hat{F}_j)\hat{T}_{i,j} + \sum_{i,j,k,l}^{N} h_i h_k \ell_C(C_{i,k}, \hat{C}_{j,l})\hat{T}_{i,j}\hat{T}_{k,l} \tag{45}$$

For the ablation study, we replace it with the one proposed to train PIGVAE [68]. Since the original PIGVAE loss cannot take into account the node padding vector $h$, we propose the following extension

$$\mathcal{L}_{\text{PIGVAE+}}(G, \hat{G}, \hat{T}) = \sum_{i}^{N} \ell_h(h_i, [\hat{T}\hat{h}]_i) + \sum_{i}^{N} h_i \ell_F(F_i, [\hat{T}\hat{F}]_i) + \sum_{i,j}^{N} h_i h_j \ell_C([C_{i,j}; \hat{T}\hat{C}\hat{T}^T]_{i,j}). \tag{46}$$

We also add a regularization term as suggested in the original paper, and extend it to take into account the padding

$$\Omega_{\text{PIGVAE+}}(\hat{T}) = -\sum_{i,j} \hat{T}_{i,j} \log(\hat{T}_{i,j}) h_j. \tag{47}$$

Finally, we replace our loss by $\mathcal{L}_{\text{PIGVAE+}}(G, \hat{G}, \hat{T}) + \lambda \Omega_{\text{PIGVAE+}}(\hat{T})$ and we report the results for $\lambda = 10$ (after a basic grid-search $\lambda \in \{0.1, 1, 10\}$).

**Featurizer.** As detailed in C, the proposed featurizer $\phi$ augments the graph representation with high-order properties such as the shortest path matrix. We check the importance of this preprocessing step by removing it entirely. More precisely, we change the equation (16) that defines the node features into

$$F(x) = F_0(x) \tag{48}$$

and the equation (17) that defines the edge features into

$$C_{i,j}(x) = \text{CONCAT}[F_i(x), F_j(x), \text{ONE-HOT}(A_{i,j}(x))] \tag{49}$$

**Encoder.** To assess the importance of the Evoformer Encoder module, we swap it with a graph neural network (GNN). More precisely, we change equation (19) into

$$F^{l+1} = \texttt{GNN}(F^l, A) \tag{50}$$

where $A$ is the adjacency matrix. Since the GNN does not output hidden edge representations, we define them as $C_{i,j}^L = \texttt{CONCAT}[F_i^L, F_j^L]$. For this experiment, we use a 4-layer GIN [73].

**Decoder.** Similarly, we check the importance of the novel Evoformer Decoder by swapping it with a more classical Transformer Decoder. More precisely, we change equation (23) into

$$F_Q^{l+1} = \texttt{TransformerDecoder}(F_Q^l, Z^L) \tag{51}$$

Since the Transformer Decoder does not reconstruct any edges, we add an extra MLP $(C_Q^L)_{i,j} = \texttt{MLP}\big(\texttt{CONCAT}[(F_Q^L)_i, (F_Q^L)_j]\big)$.

**Matcher.** Since the role of our matcher is very similar to that of the permuter introduced for PIGVAE [68], we propose to plug it inside our model instead. For completeness, we recall the definition of PIGVAE permuter:

$$m(\hat{X}, X) = \texttt{SoftSort}(XU^T) \tag{52}$$

where $U \in \mathbf{R}^d$ is learnable scoring vector and the $\texttt{SoftSort} : \mathbb{R}^N \mapsto \mathbb{R}^{N^2}$ operator is defined as the relaxation of the $\texttt{ArgSort}$ operator

$$\texttt{SoftSort}(s)_{i,j} = \texttt{softmax}\left( \frac{|s_i - \texttt{sort}(s)_j|}{\tau} \right) \tag{53}$$

where $\tau > 0$ is a fixed temperature parameter. Note that, compared to the GRALE matcher, this implementation does not leverage the node features of the output graphs. Instead, it assumes that the permutation between input and output can be seen as a sorting of some node scores $s_i$. Importantly, the original paper mentions that the Permuter benefits from decaying the parameter $\tau$ during training. However, detailed scheduling training is not provided in the original article; therefore, we report the best results from the grid search $\tau \in \{1e-5, 1e-4, 1e-3, 1e-2\}$.

**Disanbiguation noise.** Finally, we propose to remove the disambiguation noise added to the input features. That is

$$\phi(x) = (F(x), C(x)). \tag{54}$$

Recall that the expected role of this noise is to enable the model to produce distinct node embeddings for nodes that are otherwise undistinguishable (in the sense of the Weisfeiler-Lehman test).

## F Proofs of the theoretical results

### F.1 Loss properties

**Proposition 1: Computational cost.** This proposition is a trivial extension of Proposition 5 from Any2Graph [37]. The only difference is that, as proposed in [76], we consider an edge feature tensor $C$ instead of an adjacency matrix $A$. The proof remains the same. Note that the assumption made in the original paper is that there exist $h_1, h_2, f_1, f_2$ such that $\ell_C(a, b) = f_1(a) + f_2(b) - \langle h_1(a), h_2(b) \rangle$. Instead, we make the slightly stronger (but arguably simpler) assumption that $\ell_C$ is a Bregman divergence. By definition, any Bregman divergence $\ell$ writes as

$$\ell(a, b) = F(b) - F(a) - \langle \nabla F(a), b - a \rangle \tag{55}$$

and thus, the original assumption is verified for $f_1(a) = \langle \nabla F(a), a \rangle - F(a)$, $f_2(b) = F(b)$, $h_1(a) = \nabla F(a)$ and $h_2(b) = b$.

**Proposition 2: Positivity.** This is a direct extension to the case $n_C > 1$ of Proposition 3 from [37].

### F.2 Positioning with respect to PIGVAE

In the following, we assume that the ground losses $\ell_F$ and $\ell_C$ are Bregman divergences as defined above. $G = (F, C)$ and $\hat{G} = (\hat{F}, \hat{C})$ are graphs of size $N$ and we omit the padding vectors, enabling fair comparison with PIGVAE. In this context, the proposed loss is rewritten as

$$\mathcal{L}_{\text{OT}}(G, \hat{G}, \hat{T}) = \sum_{i,j}^{N} \ell_F(F_i, \hat{F}_j)\hat{T}_{i,j} + \sum_{i,j,k,l}^{N} \ell_C(C_{i,k}, \hat{C}_{j,l})\hat{T}_{i,j}\hat{T}_{k,l}. \tag{56}$$

We also recall the expression of PIGVAE's loss

$$\mathcal{L}_{\text{PIGVAE}}(G, \hat{G}, \hat{T}) = \mathcal{L}_{\text{ALIGN}}(G, \hat{T}[\hat{G}]), \tag{57}$$

where $\mathcal{L}_{\text{ALIGN}}(G, \hat{G}) = \sum_{i=1}^{N} \ell_F(F_i, \hat{F}_i) + \sum_{i,j=1}^{N} \ell_C(C_{i,j}, \hat{C}_{i,j})$ and $\hat{T}[\hat{G}] = (\hat{T}\hat{F}, \hat{T}\hat{C}\hat{T}^T)$.

**Proposition 3: Link between $\mathcal{L}_{\text{PIGVAE}}$ and $\mathcal{L}_{\text{OT}}$.** Let $\hat{T} \in \pi_N$ be a bistochastic matrix. Since $\ell_F$ is a Bregman divergence, it is convex with respect to the second variable and the Jensen inequality gives:

$$\sum_{j}^{N} \ell_F(F_i, \hat{F}_j)\hat{T}_{i,j} \geq \ell_F\left(F_i, \sum_{j}^{N} \hat{F}_j\hat{T}_{i,j}\right) = \ell_F(F_i, [\hat{T}\hat{F}]_i). \tag{58}$$

Note that Jensen's inequality applies because, by definition of a bistochastic matrix, $\sum_j T_{i,j} = 1$. Applying the same reasoning **twice** we get that for any $i, k$

$$\sum_{j,l}^{N} \ell_C(C_{i,k}, \hat{C}_{j,l})\hat{T}_{i,j}\hat{T}_{k,l} \geq \ell_C\left(C_{i,k}, \sum_{j,l}^{N} \hat{C}_{j,l}\hat{T}_{i,j}\hat{T}_{k,l}\right) = \ell_C(C_{i,k}, [\hat{T}\hat{C}\hat{T}^T]_{i,k})) \tag{59}$$

which concludes that

$$\mathcal{L}_{\text{OT}}(G, \hat{G}, \hat{T}) \geq \mathcal{L}_{\text{PIGVAE}}(G, \hat{G}, \hat{T}) \tag{60}$$

With equality if and only if all the Jensen inequalities are equalities, that is, if and only if $\hat{T}$ is a permutation matrix.

**Proposition 4: Failure case of $\mathcal{L}_{\text{PIGVAE}}$.** PIGVAE's loss can be zero even if $\hat{G}$ and $G$ are not isomorphic. This can be demonstrated with a very simple counterexample. Let $N = 2$, $C = \hat{C} = 0$ and

$$F = \begin{pmatrix} 0.5 \\ 0.5 \end{pmatrix}, \quad \hat{F} = \begin{pmatrix} 1 \\ 0 \end{pmatrix} \tag{61}$$

While it is obvious that the two graphs are not isomorphic (the sets of nodes are different), when we set the matching matrix to

$$\hat{T} = \begin{pmatrix} 0.5 & 0.5 \\ 0.5 & 0.5 \end{pmatrix} \tag{62}$$

we have that $\hat{T}\hat{F} = F$ and that $\mathcal{L}_{\text{PIGVAE}}(G, \hat{G}, \hat{T}) = 0$. Therefore, we conclude that $\mathcal{L}_{\text{PIGVAE}}$ does not satisfy proposition 2.

