# OpenReview forum: "The quest for the GRAph Level autoEncoder (GRALE)"
_NeurIPS.cc/2025/Conference — NeurIPS 2025 poster_

### Official Review · Reviewer_s35a · 2025-06-25

**Clarity:** 2
**Significance:** 2
**Originality:** 3
**Rating:** 5
**Confidence:** 3

**Summary:**

This work proposes GRALE, a graph auto-encoder (GAE) to generate representations for graph structure and node features. Graphs with varying sizes can be encoded/decoded by padding. The graph representation can be well conducted on various downstream tasks.

**Questions:**

This work proposes a graph autoencoder to generate both graph and node level embeddings for graphs with different sizes; using padding to align with the max size of  graphs. This work provides a detailed block design for the entire encoder/decoder framework and clear flow of evaluation. Generally I appreciate the technical contribution and completeness of the GRALE in this work. To save authors’ time, I may directly express my confusions and questions on GRALE:

1. The introduction of GRALE is not easy to follow for readers not focusing on the GAE area. Many concepts are not well explained and it takes me some time to understand them by reviewing the appendix and papers referred.

1.1 In the Introduction, authors claimed GAE like Graph U-Net take advantage of the ground-truth information, including the graph size, in the decoding phase. Could you give more explanation on it? Why is this not a good choice in practice? For example in Graph U-Net, the graph size/structure is utilized the information to conduct the pooling/global-pooling function, and finally we get the graph embedding for downstream tasks. It seems to me a similar approach as GRALE where “pooling” is utilized in GRALE’s decoder. Please further justify that Graph U-Net-like structures are not an appropriate one in your scenario.

1.2 Regarding the padding, “N is a maximum graph size; all graphs are padded to this size.” May I ask if the padding strategy (for example where the zeros should be padded, evenly on  two sides of the graph, $[…]+n+[…]\to N$, or simply just after the graph $n+[…]\to N$.) Will the padding detail influence the model performance?

1.3 In Sec. 3.4, the Matcher $m$ is introduced to calculate and projected into a differentiable manner for the GRALE loss later. I understand the intension of this module, but have no idea of the mechanism it uses. Might be clearer to explicitly show the expression (or quick illustration) for readers like me without knowledge of it. Also, In Sec. 4.3, authors mention that most GAE works treat matching as a regression problem, but in GRALE, the matcher is trained without groundtruth but minimizing the cost. To my understanding, is this also a regression problem? I mean the $\hat{T}=SINKHORN(K)$ seems still a regression because $K$ comes from the affinity between $X$ and $\hat{X}$, where $X$ is the processed “ground-truth”.

2. The evaluation is comprehensively conducted, but some concerns from my current understanding.

2.1 In Table 1, the GRALE outperforms other methods a lot, which is quite impressive. Besides the results, can you give some interpretation on why GRALE can have this good performance?

2.2 Quickly following, also in the main body of this manuscript, it’s much better to include more analysis and justification on the design of modules. For example in each module introduction, could you talk about why you need these operations? Why your choice is or should be the optimal one? Simply stating these might not be convincing enough.

2.3 In Sec.5.2, great qualitative evlauations! But could you provide more analysis and explanation on the results; those would be more instructive to followed researchers.

3. The limitation of the GRALE is on the max graph size. I think this stems from the large model size, as well the training effort, of the proposed GRALE? Is there some intuitive or thinking-loudly strategy could mitigate this issue?

**Minor comments:**

i. “This module first extract from data” -> “extracts”

ii. “Remplaced” in Table 2 -> “Replaced”

iii. In Figure 11, the reconstruction failure on symmetric structure seems similar to the issue mentioned in [1]. It is interesting. If possible, could you share more opinion on this? I found your manuscript (using random noise) and this paper (using two branches to generate two different embedding) seem address the same issue in two ways.

iv. The proposed new decoder in GRALE seems a good contribution. Authors might consider moving it to the main body and provide more interpretation on it.

[1] Duan, Shijin, et al. "GraphCroc: Cross-Correlation Autoencoder for Graph Structural Reconstruction." Advances in Neural Information Processing Systems 37 (2024): 45409-45437.

**Ethical Concerns:**

["NO or VERY MINOR ethics concerns only"]

**Final Justification:**

This work proposes a graph autoencoder to generate both graph and node level embeddings for graphs with different sizes; using padding to align with the max size of  graphs. In the rebuttal, the authors clearly answered the questions that I raised, also with proper references. I have no further questions. I believe with the clarification, the revision will be more fluent for readers to understand their design philosophy and module detail. Thus I increase my rating to "accept", and hope it can have valuable contribution to the graph learning community.

**Limitations:**

Please see the questions.

**Quality:**

3

**Strengths And Weaknesses:**

**Strengths**

1. Completeness: The work clearly defines the problem and outlines the modules in GRALE.

2. Technical depth: The work provides a detailed design for the entire framework.

**Weaknesses**

1. Writing/Clarity: The paper does not adequately justify or clarify some content, making it difficult to follow.

2. Novelty: While the framework design is comprehensive, its novelty should be further highlighted.

---

> ### Author Rebuttal · Authors · 2025-07-29
>
> Thank you for your feedback. You highlighted several aspects of the paper that need to be clarified. We provide the requested explanations below and will include these additional details in the final version of the paper, using the extra page available.
>
> > - Please further justify that Graph U-Net-like structures are not an appropriate one in your scenario
>
> Formally, Graph U-Net-like architectures are written as follows:
>
> $$
> g(G) = Z, I \tag{U-Net Encoder}
> $$
>
> where $G$ is the input graph, $Z$ is the latent representation (permutation invariant), and $I$ contains additional information (not permutation invariant), such as the list of nodes removed at each layer. The associated decoder is:
>
> $$
> f(Z, I) = G' \tag{U-Net Decoder}
> $$
>
> That is, the decoder uses both the latent representation and the additional information to reconstruct the graph $G'$.
>
> This structure implies that certain key properties of the graph, such as the number of nodes, are not encoded in the latent space but instead stored in $I$. As a result, the decoder cannot reconstruct a graph from an arbitrary point $Z$ in latent space without access to $I$. This limitation makes many of GRALE's intended applications infeasible (e.g., interpolation or sampling in latent space). We will make this point clearer in the final version of the paper.
>
> > - Will the padding order influence the model performance?
>
> No, it won't. Thank you for the question—this is an important feature of GRALE. Because our loss is permutation invariant, any reordering of the input graph (including padded nodes) has no effect on the output. In particular, whether padded nodes appear at the beginning, the end, or in random positions has no impact on (a) the latent representation, or (b) the final loss.
>
> > - In Sec. 3.4, the Matcher $m$ is introduced to calculate and projected into a differentiable manner for the GRALE loss later. I understand the intension of this module, but have no idea of the mechanism it uses. Might be clearer to explicitly show the expression (or quick illustration) for readers like me without knowledge of it. Also, In Sec. 4.3, authors mention that most GAE works treat matching as a regression problem, but in GRALE, the matcher is trained without groundtruth but minimizing the cost. To my understanding, is this also a regression problem? I mean the $\hat{T} = \text{SINKHORN}(K)$ seems still a regression because comes from the affinity between $X$ and $\hat{X}$, where is the processed “ground-truth”.
>
> Thank you—this is an important comment. Indeed, several clarifications were missing.
>
> The core intuition is as follows: **graph matching between $G$ and $\hat{G}$ is hard, but we aim to learn node embeddings ($X$ and $\hat{X}$) such that the optimal matching between the embeddings (which is easier to compute) approximates the matching between the original graphs.**
>
> With this goal in mind, we use tools from Optimal Transport (see [1] for a detailed introduction).
>
> The optimal matching between two sets of nodes (no edges) can be written as:
>
> $$
> \hat{P} = \arg\min_{P \in \pi_N} \sum_{i,j} P_{i,j} \, d(X_i,\hat{X}_j) \tag{OT}
> $$
>
> where $d$ is a distance between node embeddings. However, this solution is non-differentiable with respect to $X$ and $\hat{X}$, which prevents backpropagation.
>
> To address this, we use the entropic regularization of the problem:
>
> $$
> \hat{T} = \arg\min_{T \in \pi_N} \sum_{i,j} T_{i,j} \, d(X_i,\hat{X}_j) + \epsilon H(T) \tag{Sinkhorn}
> $$
>
> where $H(T)$ is the negative entropy of $T$. This formulation is differentiable and thus suitable for end-to-end training [2].
>
> Moreover, the solution $\hat{T}$ can be computed as:
>
> $$
> \hat{T} = \texttt{BREGMAN}(K)
> $$
>
> where $K_{i,j} = \exp\left(-\frac{d(X_i,\hat{X}_j)}{\epsilon}\right)$, and $\texttt{BREGMAN}$ is the Bregman algorithm: it alternates row and column normalizations of $K$ until convergence. We use 100 iterations, making $\texttt{BREGMAN}$ a differentiable module with respect to $X$ and $\hat{X}$.
>
> Rather than choosing a fixed distance $d$ and regularization coefficient $\epsilon$, we let the model learn both via two MLPs:
>
> $$
> \frac{d(X_i, X_j)}{\epsilon}=\left\|  \texttt{MLP}\_\text{in}(X_i) - \texttt{MLP}\_\text{out}(X_j)\right|
> $$
>
> Finally, let us clarify the claim that we train "without ground truth". Recall that the goal of the matcher is to approximate the optimal matching between $G$ and $G'$:
>
> \begin{equation}
>     T^* = \text{arg}\,\min\limits_{T \in \pi_N} \mathcal{L}_{OT}(G, G',T)
> \end{equation}
>
> A supervised regression based approach would train following a supervised loss such as $||\hat{T} - T^* ||_2$, but this approach requires to compute the target $T^*$ which is NP-hard. instead we propose to directly minimize
>
> \begin{equation}
>      \mathcal{L}_{OT}(G, G',\hat{T})
> \end{equation}
>
> This makes sense because $\mathcal{L}\_{OT}(G, G',\hat{T}) \geq \mathcal{L}\_{OT}(G, G',T^*)$ and thus we indirectly minimize the otherwise untrackable loss.
>
> We hope that this extra set of information clarify the proposed matcher. We fully agree that these informations should be provided in the main body of the paper and will use the additionnal page for this. Thank you again for raising this point.
>
> > - In Table 1, the GRALE outperforms other methods a lot, which is quite impressive. Besides the results, can you give some interpretation on why GRALE can have this good performance
>
> We believe that the success of GRALE is ultimately a combination of factors. Our ablation studies show that each module contributes significantly to performance, making it difficult to isolate a single dominant factor. On a higher level, we think that the true explanation for this success is the strategy that we followed to design GRALE (see next answer).
>
> > - 2.2 Quickly following, also in the main body of this manuscript, it’s much better to include more analysis and justification on the design of modules. For example in each module introduction, could you talk about why you need these operations? Why your choice is or should be the optimal one? Simply stating these might not be convincing enough.
>
> We fully agree with this. We think that the reader might benefit from more insights into GRALE's conception. We will add this discussion to the final version of the paper.
>
> Broadly, our design followed a two-step process: (1) we defined a “wish-list” of desired properties for each component (see Section 2.1), and (2) we surveyed the literature to find or adapt modules that meet these requirements. For example, Evoformer was a perfect fit for our encoder: it is graph-attention-based, scalable to large datasets, and permutation equivariant. A more complex case was that of the loss: the losses proposed in PIGVAE and Any2Graph were very close to our aim but were still missing a few components so we adapted them. No suitable decoder existed, so we proposed a new one inspired by both the Evoformer encoder and Transformer decoder—hence the name “Evoformer Decoder.”.
>
> Beyond the wordplay, this is why we called our work "The Quest for the Graph AutoEncoder".
>
> > - 2.2 Quickly following, also in the main body of this manuscript, it’s much better to include more analysis and justification on the design of modules. For example in each module introduction, could you talk about why you need these operations? Why your choice is or should be the optimal one? Simply stating these might not be convincing enough.
>
> We fully agree. In the final version, we will include more discussion and justification for each module, as illustrated in the reply above.
>
> > - The proposed new decoder in GRALE seems a good contribution. Authors might consider moving it to the main body and provide more interpretation on it.
>
> Thank you for this very positive feedback. This is our belief as well. We think that the clarifications that you suggested are also very important, but if we have some room left we will definitely move it to the main body of the paper. Note that as suggested by the other reviewers Figure 4 might go the appendices. With the additionnal page of the final version, we think we have the room for everything.
>
> **References**
>
> [1] Peyré, G., & Cuturi, M. (2019). *Computational Optimal Transport: With Applications to Data Science*. Foundations and Trends® in Machine Learning, 11(5-6), 355–607.
> [2] Genevay, A., Peyré, G., & Cuturi, M. (2018). *Learning generative models with Sinkhorn divergences*. In AISTATS, pp. 1608–1617.

---

> > ### Comment · Reviewer_s35a · 2025-08-03
> >
> > Thank you for the detailed clarification. I have no further questions. I hope it makes a valuable contribution to the graph community.

---

### Official Review · Reviewer_XSRE · 2025-06-29

**Clarity:** 4
**Significance:** 4
**Originality:** 3
**Rating:** 5
**Confidence:** 4

**Summary:**

A fully differentiable graph autoencoder is introduced, including a suggestion of how to make node matching differentiable.

**Questions:**

Figure 4 seems cherry picked to me. All three properties scale with the graph size, as molecular weight and solubility are correlated, and also u0 as an extensive quantity is highly dependent on the graph size. It is not surprising that the embedding also scales with the graph size, as the decoder hast to reconstruct the mask, i.e. number of nodes from the latent embedding. How do these plots look like if you visualize intensive quantities, that are independent of the graph size (e.g. HOMO/LUMO energies or other intensive quantities in QM9). Showing plots of all QM9 properties in the appendix would be benefitial to support the claims in Section 5.2 or also to limit the applicability of those claims.

The same is true for Table 3 and Table 4(QM9): Which task was performed (there are many labels in QM9) for property prediction and graph prediction?

What is the computational cost of training the model compared to previous models used in the benchmark? Please add that information to the paper. Please also discuss if the training scales cubic or even more than cubic with the typical graph size in the datasets (single forward/backward pass scales probably cubic, but if more data and/or more epochs are needed for training on larger graphs, one might expect above-cubic scaling).

**Ethical Concerns:**

["NO or VERY MINOR ethics concerns only"]

**Final Justification:**

The paper is an important step in the direction of having graph autoencoders with graph neural networks also on the decoder side, rather than decoders that generate other graph representations in an indirect way. The method presented here is sufficiently different from other recent publications, and the approach is interesting, even though it has some disadvantages, such as the scaling in graph size. The questions were sufficiently addressed in the rebuttal.

**Limitations:**

yes

**Paper Formatting Concerns:**

-

**Quality:**

4

**Strengths And Weaknesses:**

The paper is very clearly and well written.
It presents one of the first solutions to a long-standing problem.
The results are very good and convincing.
Even though there is related work (PIGVAE), the authors clearly discuss differences and show significant empirical advantages.
The approach is original and combines multiple known as well as new aspects in an interesting way.
The approach presented establishes a significant step forward in self-supervised graph representation learning.
The computational cost associated to the cubic scaling might become a bottleneck and should be discussed more openly in the paper.

---

> ### Author Rebuttal · Authors · 2025-07-29
>
> Thank you very much for your review. In particular, the insights you provided about chemistry were really helpful. Below, we directly address the points you raised:
>
> > Figure 4 seems cherry-picked to me. All three properties scale with the graph size, as molecular weight and solubility are correlated, and also u0 as an extensive quantity is highly dependent on the graph size. It is not surprising that the embedding also scales with the graph size, as the decoder has to reconstruct the mask, i.e., number of nodes, from the latent embedding. How do these plots look like if you visualize intensive quantities that are independent of the graph size (e.g., HOMO/LUMO energies or other intensive quantities in QM9)? Showing plots of all QM9 properties in the appendix would be beneficial...
>
> We agree that it is not surprising to observe graph size in the latent space, since one of the objectives of the model is to reconstruct graphs of varying sizes. This was more of a sanity check than a key result. That said, we still find it interesting that this appears so clearly — and linearly — in the first two PCA components.
>
> However, we were unaware that u0 and solubility were strongly correlated with the number of atoms. Thank you for pointing this out. For context, we selected u0 because it was commonly used as a regression benchmark on the QM9 dataset (out of the 17 targets, it was the only one tracked on the PapersWithCode leaderboard at that time). We selected solubility because it could be computed using RDKit and seemed like a chemically meaningful target for otherwise unsupervised PUBCHEM dataset.
>
> Following your suggestion, we applied the same visualization technique to intensive properties like HOMO/LUMO energies. These required additional tuning of the t-SNE parameters to obtain meaningful 2D visualizations.
>
> Thanks to your comment, we now recognize that this figure is less informative than we originally thought. We are considering moving it to the appendix, along with the new HOMO/LUMO plots, and will include a more detailed discussion.
>
> > The same is true for Table 3 and Table 4 (QM9): Which task was performed (there are many labels in QM9) for property prediction and graph prediction?
>
> You're absolutely right — this information was missing. We will add it to Table 7. Here is the missing column:
>
> | Dataset | Target |
> |---------|--------|
> | QM9     | internal energy (u0) |
> | QM40    | internal energy (u0) |
> | ESOL    | water solubility |
> | LIPO    | octanol/water distribution coefficient |
> | FREESOLV | hydration free energy of small molecules |
> | BBBP    | binary labels for blood-brain barrier permeability |
> | BACE    | IC50 (bioactivity classification) |
>
> Note that for all datasets we used the regression/classification targets defined in the MoleculeNet benchmark [1]. The only exception is QM9 and QM40, where we reported u0 for the reasons mentioned above.
>
> As for Table 4, it corresponds to a different type of task: **supervised graph prediction**. Here, the goal is to predict an entire graph given some input $x$. In particular, we reproduced the benchmark from the Any2Graph paper, which includes, for instance, reconstructing molecular graphs from their Morgan radius-2 fingerprints ($x$ is the fingerprint). In this table, the "QM9" column refers to such tasks.
>
> > What is the computational cost of training the model compared to previous models used in the benchmark? Please add that information to the paper.
>
> This was also noted by reviewer ‘fwJt’, this information is missing. We will add the following comparison. To ensure fairness, we implemented all models using the same architecture (e.g., Evoformer as encoder). This ensures fair accuracy comparison, though simpler architectures could make some models more time efficient (e.g., GVAE) at the cost of reduced performance.
>
> | Model    | Training Time (COLORING) | Training Time (PUBCHEM 16) | Training Time (PUBCHEM 32) | Loss Complexity      |
> |----------|---------------------------|------------------------------|------------------------------|------------------------|
> | GVAE     | 8h                        | 18h                          | 90h                          | $\mathcal{O}(N^2)$     |
> | GraphVAE | 40h                       | 80h                          | N.A.                         | $\mathcal{O}(N^4)$     |
> | PIGVAE   | 8h                        | 18h                          | 95h                          | $\mathcal{O}(N^3)$     |
> | GRALE    | 8h                        | 20h                          | 100h                         | $\mathcal{O}(N^3)$     |
>
> Given the same architecture and hardware (as reported in table 9), most methods have comparable training costs. The exception is GraphVAE whose $\mathcal{O}(N^4)$ loss introduces significant overhead.
>
> > Please also discuss if the training scales cubic or even more than cubic with the typical graph size in the datasets (a single forward/backward pass scales probably cubic, but if more data and/or more epochs are needed, one might expect above-cubic scaling).
>
> This is a great point, and we will include this discussion. As discussed with Reviewer LtMM, while our current implementation is $\mathcal{O}(N^3)$, it would be relatively easy to design an architecture with $\mathcal{O}(N^2)$ complexity (see baselines in Appendix E). However, the true bottleneck is the loss, which remains $\mathcal{O}(N^3)$ even assuming an oracle matcher (Proposition 1). To reduce this, approximating the loss (e.g., via low-rank approximations [2]) would be a promising direction. This could reduce reconstruction quality but might still result in a rich latent space.
>
> As for **total training time**, you raise an interesting point. When moving from PUBCHEM 16 to PUBCHEM 32, we needed to double the number of gradient steps to reach convergence. It’s hard to determine whether this is due to the increased graph size or the larger dataset size (PUBCHEM 32 is six times larger, to be exact). Disentangling these effects would require careful and large-scale experiments, which are beyond the scope of this paper — but this is definitely worth exploring further, and we will add it to the discussion section.
>
> ---
>
> ### References
>
> [1] Wu, Z., Ramsundar, B., Feinberg, E. N., Gomes, J., Geniesse, C., Pappu, A. S., ... & Pande, V. (2018). MoleculeNet: A benchmark for molecular machine learning. *Chemical Science*, 9(2), 513–530.
>
> [2] Scetbon, M., Peyré, G., & Cuturi, M. (2022). Linear-time Gromov-Wasserstein distances using low-rank couplings and costs. *PMLR*.

---

> > ### Comment · Reviewer_XSRE · 2025-08-02
> >
> > Thank you very much for your detailed answers. I am very interested to see the latent space structure for non-extensive properties, and if they are less conclusive than U0/solubility ordering, then I support moving them to the appendix.
> >
> > Thank you also for your compute time tests.
> >
> > I am convinced that this is a very valuable contribution, and I stand behind my "accept" rating.

---

### Official Review · Reviewer_fwJt · 2025-07-01

**Clarity:** 3
**Significance:** 3
**Originality:** 3
**Rating:** 5
**Confidence:** 3

**Summary:**

The paper proposes a graph autoencoder that works at the graph level (ideally without directly using node embeddings, although in practice this is not entirely true). The main idea is to avoid the expensive matching step between the input graph (of the encoder) and the output graph (from the decoder) by using a loss inspired by Optimal Transport. Instead of computing a permutation matrix, the method computes a transport matrix. Since this operation can also be complex, the model includes a module that learns to predict the matrix. The whole system is trained end-to-end.

**Questions:**

- How much of the performance depends on the preprocessing stage? This aspect is not clearly discussed and might deserve more attention
- It’s not clear to me how the maximum number of nodes is fixed.

For other questions, please see above.

**Ethical Concerns:**

["NO or VERY MINOR ethics concerns only"]

**Final Justification:**

After considering the rebuttal, I maintain my positive assessment of the paper. The authors provided clear answers to most of my questions, such as clarifying the relationship between graph-level and node-level embeddings, explaining the K-token representation in more detail, and adding the missing training time comparison. My positive recommendation is supported by 1) an elegant end-to-end approach, 2) valuable clarifications in the rebuttal, and 3) competitive results, even though there is still room for improvement.

**Limitations:**

-

**Paper Formatting Concerns:**

Typos  >>> PIGVAE architecture is composed of 3 main blocs:

**Quality:**

3

**Strengths And Weaknesses:**

*Strengths*
- The paper is well structured and easy to follow; the transition between (almost all) sections is smooth and helps the reader follow the flow of ideas.
- The choice of using different schemes to build the input and target graphs is interesting, as well as highlighting that breaking the symmetry between input and output can improve learning.
- The entire system, including the prediction of the graph matching, is trained end-to-end, which makes the approach elegant and potentially efficient.

*Weaknesses*

- The paper claims to produce a graph-level embedding, but in reality, it also produces node embeddings.
- The paper states that it produces a graph-level embedding Z, but in practice, it is a representation composed of K tokens. This is an important point, but it is only briefly discussed. What does it mean that these tokens represent abstract "concepts"?
- Figure 2 could be very helpful, but in its current form, it is hard to read. In particular, it is unclear where X comes from, what the "linear" block actually does, and what the K \times D input to the transformer decoder represents (or maybe I missed some parts). The best solution would be to expand the figure caption to clearly describe the architecture, including a proper explanation of notations such as n_F, n_C, etc.
- The readability of Figure 3 could be improved, for instance, by adding the labels K and D.
- Figure 4 does not provide much insight. It would be more informative to include a comparison with existing methods (e.g., PIGVAE) to better assess the qualitative performance of GRALE.
- Unless I missed it, there seems to be no table reporting training time comparisons with other methods.
- While the paper emphasizes the importance of the proposed matching-free loss, the ablation results in Table 2 suggest that its contribution to reconstruction quality is relatively limited compared to other architectural components such as the encoder, matcher, or featurizer. This raises a question: could this be considered a discrepancy? In other words, are the empirical differences between the proposed loss and previous alternatives negligible in practice? Apart from the observation that GRALE shows reduced standard deviation (suggesting more stable training), the actual impact of the loss function seems limited, and it is not entirely clear how the authors interpret this result.
- Complex graph operations in the latent space are presented as an ambitious aspect of the paper, but they are only briefly introduced. Graph interpolation in latent space is not a novel idea; actually, it is well established in the graph generative modeling community. Its effectiveness depends on how well-structured the latent space is (ie embeddings must reflect meaningful semantics, where similar graphs correspond to nearby vectors), otherwise, interpolations may lead to invalid or meaningless graphs. In this work, the discussion remains theoretical, making this part of the contribution speculative.

---

> ### Author Rebuttal · Authors · 2025-07-29
>
> Thank you for your positive and in-depth review. In particular, we really appreciate the feedback on the figures; we will update them accordingly. We address your other questions below:
>
> > The paper claims to produce a graph-level embedding, but in reality, it also produces node embeddings.
>
> That’s true, and it’s an aspect of the paper that deserves more discussion.
>
> Ultimately, we obtain both a graph-level embedding that is permutation invariant, and a set of node-level embeddings that are permutation equivariant. However, it’s very important to emphasize that **all the information required to reconstruct the graph is stored in the graph-level embedding**. The node embeddings are only passed to the matcher, which outputs a permutation matrix used solely for loss computation. Thus, the model cannot “cheat” by encoding structure in the node embeddings — the graph-level embedding acts as a true bottleneck, as in a standard autoencoder.
>
> From a more abstract perspective, this means that our model naturally disentangles the permutation-invariant part of the input (stored in the graph-level embedding) from the arbitrary node ordering (stored in the node-level embeddings). We recently noticed a connection with similar strategies for disentangling other invariances, such as rotation symmetry [2]. We will add this discussion to the final version of the paper.
>
> Overall, we believe this is a strength rather than a weakness. In particular, the node-level embeddings prove useful for downstream tasks, as demonstrated in the graph matching experiment (see Table 5, that was further improved via fine-tuning thanks to Reviewer LtMM’s suggestion).
>
> > The paper states that it produces a graph-level embedding Z, but in practice, it is a representation composed of K tokens. This is an important point, but it is only briefly discussed. What does it mean that these tokens represent abstract "concepts"?
>
> That's a fair point — the word “concepts” is too strong and implies a level of interpretability we do not yet justify. We will remove this term in the final version.
>
> What we can say without additional experiments is that (1) these tokens encode higher-order properties of the graph, as they result from cross-attention over all node-level embeddings, and (2) this token-based representation is empirically effective, as shown in Figure 3. As for why this works well, our hypothesis (mentioned in line 231) is that it aligns better with attention-based architectures — though more subtle mechanisms may be at play. A deeper analysis is out of scope for this paper.
>
> > Figure 4 does not provide much insight. It would be more informative to include a comparison with existing methods (e.g., PIGVAE) to better assess the qualitative performance of GRALE.
>
> Following your remark and a similar one from Reviewer XSRE, we realized that this figure is less informative than we initially thought. We are considering moving it to the appendix and adding a new plot (as you suggested) to enable a more meaningful qualitative comparison.
>
> > Unless I missed it, there seems to be no table reporting training time comparisons with other methods.
>
> You're right — this table was missing. We provide it below and will include it in the final version. To ensure a fair comparison, all methods were implemented with the same architecture (e.g., using Evoformer as the encoder). This improves accuracy comparability, though some methods like GVAE could potentially be made more time efficient with a simpler architecture at the cost of reduced performance.
>
> | Model    | Training Time (COLORING) | Training Time (PUBCHEM 16) | Training Time (PUBCHEM 32) | Loss Complexity      |
> |----------|---------------------------|------------------------------|------------------------------|------------------------|
> | GVAE     | 8h                        | 18h                          | 90h                          | $\mathcal{O}(N^2)$     |
> | GraphVAE | 40h                       | 80h                          | N.A.                         | $\mathcal{O}(N^4)$     |
> | PIGVAE   | 8h                        | 18h                          | 95h                          | $\mathcal{O}(N^3)$     |
> | GRALE    | 8h                        | 20h                          | 100h                         | $\mathcal{O}(N^3)$     |
>
> Given the same architecture and hardware (as reported in table 9), most methods have comparable training costs. The exception is GraphVAE, whose $\mathcal{O}(N^4)$ loss introduces significant overhead.
>
> > While the paper emphasizes the importance of the proposed matching-free loss, the ablation results in Table 2 suggest that its contribution to reconstruction quality is relatively limited compared to other architectural components.
>
> Thank you for pointing that out. In the current version (Appendix E), we compare our proposed loss to a strong baseline — the PIGVAE loss — which includes an additional regularization term, and for which we performed a grid search to optimize its weight. Despite this, our loss (without additional parameters) slightly outperforms it on average, which we consider a significant result.
>
> Moreover, as you noted, the lower standard deviation suggests greater training stability. Empirically, we observed that $\mathcal{L}_\text{PIGVAE}$ often gets stuck in local minima. This aligns with Proposition 4: the PIGVAE loss is highly sensitive to the balance between the reconstruction and regularization terms. Our proposed loss bypasses this issue entirely.
>
> > How much of the performance depends on the preprocessing stage? This aspect is not clearly discussed and might deserve more attention.
>
> We use very simple preprocessing. In particular, for molecular data, we do not incorporate domain-specific knowledge — though doing so would likely improve performance. As detailed in Appendix C.1, our preprocessing simply involves adding higher-order interactions to the graph features and edges (via powers of the adjacency matrix and feature diffusion).
>
> This strategy was tested in the ablation studies and showed a non-negligible impact (see the second line of Table 2). We concede that this is not clearly presented without consulting the appendix, and we will improve this in the final version.
>
> > It’s not clear to me how the maximum number of nodes is fixed.
>
> Thanks for pointing this out — we forgot to include this simple detail. We set the model's maximum graph size to match the largest graph in the dataset. In principle, this could be set higher (i.e., overparameterized), but we chose not to explore that due to the associated computational costs.
>
> ---
>
> ### References
>
> [1] Zhang, M., Jiang, S., Cui, Z., Garnett, R., & Chen, Y. (2019). D-VAE: A variational autoencoder for directed acyclic graphs. *NeurIPS*.
>
> [2] Katzir, O., Lischinski, D., & Cohen-Or, D. (2022). Shape-pose disentanglement using SE(3)-equivariant vector neurons. *ECCV*.

---

> ### Comment · Reviewer_fwJt · 2025-08-08
>
> I thank the authors for all the clear explanations.

---

### Official Review · Reviewer_CWAG · 2025-07-03

**Clarity:** 3
**Significance:** 3
**Originality:** 3
**Rating:** 4
**Confidence:** 3

**Summary:**

This paper proposes GRALE, a graph-level autoencoder that learns to encode and decode entire graphs into a shared embedding space using an optimal transport inspired loss and a differentiable matching module. It leverages Evoformer-based architectures to support flexible graph reconstruction and shows strong results on reconstruction, classification, and prediction tasks.

**Questions:**

1. The paper claims that the graph size is predicted via the padding mask vector, yet the decoding process is not clearly explained. How does the model ensure the correct number of nodes is generated without explicit supervision on graph size?

2. Lacks ablation against standard Transformer decoders. How critical is the Evoformer Decoder to performance, and what unique capabilities does it provide?

**Ethical Concerns:**

["NO or VERY MINOR ethics concerns only"]

**Final Justification:**

1. They have expanded our evaluation with experiments on non-chemical datasets, demonstrating the method’s generalization beyond its original domain.
2. They conducted an ablation study to validate the encoder’s effectiveness.
3. Their response addresses my concern.

**Limitations:**

1.The method incurs relatively high computational overhead, primarily due to the use of Evoformer modules and Sinkhorn-based matching, which limits scalability to larger graphs.

2. Although the OT-inspired loss is conceptually sound, the paper lacks theoretical analysis to guarantee that the learned embeddings faithfully preserve semantic or structural similarities between graphs.

**Quality:**

3

**Strengths And Weaknesses:**

Strengths:

1. GRALE embeds graphs into a tokenized vector space that supports a wide range of downstream tasks. The design supports vector arithmetic over graph embeddings, enabling novel use cases such as latent space editing and interpolation.

2 . The method achieves superior performance over GraphVAE and PIGVAE in reconstruction accuracy and downstream tasks. Ablation studies are well-conducted and demonstrate the contribution of each module.

 3.The incorporation of the Evoformer module from AlphaFold for both graph encoding and decoding is a noteworthy attempt, which might help effective modeling of interactions between node and edge features.


Weaknesses:
1. The paper omits some relevant graph-level models that could serve as strong baselines or comparisons. For instance, recent masked graph autoencoders (e.g., GraphMAE) and Transformer-based architectures for graphs (e.g., Graphormer, TokenGT) are not discussed or evaluated.

2 . The proposed “Evoformer decoder” part lacks comparative analysis with standard Transformer decoders or other graph generation decoders.

3. Most experiments are performed on molecules or COLORING datasets. It remains unclear whether the method generalizes well to non-chemical graphs

---

> ### Author Rebuttal · Authors · 2025-07-30
>
> Thank you for your review. You raised important points that motivated us to conduct new experiments. We believe these new results strengthen the paper, and we hope you agree.
>
> > Most experiments are performed on molecules or COLORING datasets. It remains unclear whether the method generalizes well to non-chemical graphs.
>
> We agree. GRALE was designed as a general graph representation learning approach, and it is indeed interesting to evaluate it on a wider range of data. After reviewing the literature, we identified a very different modality: the BN dataset [1], which consists of 200,000 8-node Bayesian networks. Each network is associated with a BIC score on the Asia dataset. The task is to predict this BIC score from the graph.
>
> We pre-trained GRALE on BN in an unsupervised fashion and used the learned graph-level embeddings as input to an MLP for the downstream regression task. We were pleased to observe that this direct use of GRALE outperformed the state-of-the-art results reported in [1], including autoencoders specifically designed for this task.
>
> We reproduce the result table from [1] below, now including GRALE:
>
> | **Method** | **RMSE ↓** | **Pearson’s *r* ↑** |
> |-----------|------------|---------------------|
> | GRALE     | 0.18       | 0.98                |
> | D-VAE     | 0.30       | 0.95                |
> | S-VAE     | 0.36       | 0.93                |
> | GraphRNN  | 0.77       | 0.64                |
> | GCN       | 0.55       | 0.83                |
> | DeepGMG   | 0.78       | 0.62                |
>
> Despite the modality being very different from molecules, GRALE achieves state-of-the-art results. Thank you for raising this point — it led us to reinforce the claims of our paper.
>
> > The paper omits some relevant graph-level models that could serve as strong baselines or comparisons. For instance, recent masked graph autoencoders (e.g., GraphMAE) and Transformer-based architectures for graphs (e.g., Graphormer, TokenGT) are not discussed or evaluated.
>
> We agree that GraphMAE is a relevant and strong baseline. We performed new experiments using GraphMAE, following the same protocol as our existing baselines. While GraphMAE supports transfer learning (as we do with pre-training on PUBCHEM), it does not provide default parameters for this setting. We therefore adapted the parameters provided for MUTAG and performed a grid search over the number of training epochs (20, 30, 50) and hidden dimensions (32, 64, 128, 256) to better utilize the large pre-training dataset.
>
> The regression results are summarized below:
>
> | **Model** | **QM9** | **QM40** | **ESOL** | **LIPO** | **FreeSolv** |
> |----------|---------|----------|----------|----------|--------------|
> | **GraphMAE** (MAE ↓) | 0.222 ± 0.016 | 0.247 ± 0.036 | 0.291 ± 0.012 |  0.527 ± 0.029  |  0.378 ± 0.028  |
> | **GRALE**  (MAE ↓)    | 0.015 ± 0.001 | 0.018 ± 0.003 | 0.274 ± 0.014 | 0.511 ± 0.022 | 0.272 ± 0.017 |
>
> And the classification benchmarks:
>
> | **Model** | **BBBP** | **BACE** |
> |----------|----------|----------|
> | **GraphMAE** (ROC AUC ↑) | 0.773 ± 0.036 | 0.857 ± 0.039 |
> | **GRALE** (ROC AUC ↑)    | 0.731 ± 0.025 | 0.821 ± 0.051
>
> Overall, we find that while GraphMAE is a strong baseline, GRALE outperforms it on 5 out of 7 tasks.
>
> Regarding Transformer-based architectures, we agree they are valid encoder options. Our only claim is that attention-based architectures are more adapted to GRALE than classical GNNs — a hypothesis we test in the ablations. While many alternatives exist, benchmarking all of them is outside our scope. We will include this discussion in the final version and reference works that investigate this space more thoroughly.
>
> > The paper claims that the graph size is predicted via the padding mask vector, yet the decoding process is not clearly explained. How does the model ensure the correct number of nodes is generated without explicit supervision on graph size?
>
> We concede that this may be unclear. We will clarify it in the paper. The key lies in the first term of the training loss (Equation 2):
>
> $$
> \mathcal{L}\_{OT}(h,\hat{h},T) = \sum\_{i,j} \ell_h(h_i,\hat{h}_j)T\_{i,j}
> $$
>
> For the sake of simplicity let's assume that the matching is the identity $T = I_n$, in that case
>
> $$
> \mathcal{L}\_{OT}(h,\hat{h},I_n) = \sum_{i} \ell_h(h_i,\hat{h}_i)
> $$
>
> Here, $\hat{h}$ is the predicted mask, and $h$ is the ground truth. This loss is zero only when $h = \hat{h}$ (up to permutation $T$).
> Since the size of the graph is encoded in $h$ ($\text{size} = \sum_i h_i$) this means that we explicitly train the model to predict the correct size. Note that we use binary cross-entropy as $\ell_h$, so $\hat{h}_i$ prediction becomes a binary classification (is node $i$ active or not). Hence the decoding step is a simple treshold:
>
> $$
> h_i \leftarrow 1[h_i > 0.5]
> $$
>
> > Lacks ablation against standard Transformer decoders. How critical is the Evoformer Decoder to performance, and what unique capabilities does it provide?
>
> We did include an ablation with a standard Transformer decoder (Table 2, line 4), which increases the edit distance by a factor of 7. This highlights the importance of Evoformer in our model.
>
> Note that by construction the standard Transformer Decoder can only process a feature matrix $X \in R^{N \times d}$, therefore we had to add a MLP head to output an adjacency matrix $A_{i,j} = MLP(X_i + X_j)$. In constrast, the proposed Evoformer Decoder is (to the best of our knowledge) the first attention module that can directly reconstruct a graph (feature and adjacency matrix) from a latent space.
>
> > The method incurs relatively high computational overhead, primarily due to the use of Evoformer modules and Sinkhorn-based matching, which limits scalability to larger graphs.
>
> First of all we agree that the $\mathcal{O}(N^3)$ scaling is a limitation. There are also two aspects that we would like to highlight:
>
> 1. **Potential for Optimization**: We believe that a $\mathcal{O}(N^2)$ version of GRALE is feasible. This is straightforward for the architecture (see Appendix E), but the main bottleneck is the loss itself. Even assuming an oracle matcher (Proposition 1), the loss remains $\mathcal{O}(N^3)$. Therefore, we think that the key for reaching $\mathcal{O}(N^2)$ would be to use approximation of the loss, for instance low-rank approximation [2]. This would decrease the reconstruction quality but might still results in a rich latent space.
>
> 2. **Practical Scalability**: We want to clarify that $N=32$ is not a hard limit. Below, we provide the time/memory cost of running one gradient step (with a batch size of 1) using a RTX 3500 Ada (12GB of memory).  For $N = 128$, the memory use is 3GB. This means that 8×L40 GPUs (48GB each) could handle a batch size of 128 — a realistic setting. The model used here matches the one used for PUBCHEM.
>
> | **N**  | **8** | **16** | **32** | **64** | **128** | **256** |
> |--------|------|-------|-------|-------|--------|--------|
> | Step Time (s) | 0.01 | 0.01  | 0.02  | 0.02  | 0.03   | 0.07   |
> | GPU Memory (GB) | 0.2  | 0.3   | 0.4   | 0.8   | 3.0    | 10.7   |
>
> This supports our claim that larger graphs are feasible — similar to how AlphaFold managed long protein sequences despite similar complexity.
>
> ---
>
> **References**
>
> [1] Zhang, M., Jiang, S., Cui, Z., Garnett, R., & Chen, Y. (2019). D-VAE: A variational autoencoder for directed acyclic graphs. *NeurIPS*.
>
> [2] Scetbon, M., Peyré, G., & Cuturi, M. (2022). Linear-time Gromov Wasserstein distances using low rank couplings and costs. *ICML*.

---

> > ### Comment · Reviewer_CWAG · 2025-08-05
> > **Comments:**
> >
> > 1. Thank you for conducting the additional experiments; they’ve significantly clarified the model’s performance. I recommend incorporating these results into the main text. Moreover, since MoleculeNet comprises several datasets, but you’ve only evaluated two, I encourage you to extend your analysis to the remaining datasets as well.
> >
> > 2. Your other responses have satisfactorily addressed my concerns, and I have no further questions.

---

> > > ### Author Response · Authors · 2025-08-06
> > >
> > > Thank you for taking the time to read our rebuttal. We are happy to know that your concerns have been adressed. We agree that the new results should be added to the paper. This will be done.
> > >
> > > Regarding MoleculeNet we would like to point out that we already provide the results for 7 datasets, not two (QM9, Q40, ESOL, LIPO, FREESOLVE, BBBP and BACE). For the sake of simplicity we focused on datasets with only one regression/classification target.
> > >
> > > Once again, thank you for your important feedback!

---

### Official Review · Reviewer_LtMM · 2025-07-06

**Clarity:** 3
**Significance:** 3
**Originality:** 3
**Rating:** 4
**Confidence:** 4

**Summary:**

The paper introduces GRALE, a novel graph-level autoencoder that encodes and decodes entire graphs—of variable sizes—into a shared Euclidean latent space. GRALE employs a loss inspired by Optimal Transport and features a differentiable node-matching module, trained end-to-end with the encoder and decoder. It builds upon the Evoformer architecture from AlphaFold, adapting it for both encoding and decoding graphs. Extensive experiments show that GRALE achieves state-of-the-art performance in tasks including graph reconstruction, classification, regression, interpolation, and editing, especially in molecular and synthetic datasets.

**Questions:**

- Can GRALE scale to significantly larger graphs (e.g., protein networks or social graphs with hundreds of nodes), and what would be required to do so?

- Would incorporating graph-specific domain priors (e.g., valency in molecular graphs) improve GRALE’s reconstructions?

- Can the Evoformer Decoder be applied independently for graph generation tasks?

- Could fine-tuning the matcher module separately improve matching accuracy for unseen graph pairs?

- Is there a way to adapt GRALE to support dynamic or evolving graphs (e.g., temporal sequences)?

- How about comparing GRALE with GraphsGPT[1]?

[1] A Graph is Worth K Words: Euclideanizing Graph using Pure Transformer

**Ethical Concerns:**

["NO or VERY MINOR ethics concerns only"]

**Limitations:**

- Scalability: The O(N³) cost for attention and matching restricts practical use to small graphs (≤32 nodes).

- Data-Hungry Training: Requires very large training sets to reach high accuracy—less suitable for low-resource domains.

- No Ground Truth Matching Supervision: Matching is learned implicitly, which may limit interpretability and performance on exact alignment tasks.

- Fixed Graph Size Ceiling: Padding strategy introduces a hard constraint on maximum graph size.

- Generalization to Unstructured Inputs: Limited discussion or experiments on how well GRALE performs with real-world, noisy graph inputs outside curated datasets.

**Quality:**

3

**Strengths And Weaknesses:**

Strengths：
- Novel Architecture: Utilizes and adapts Evoformer for graph-level encoding and introduces a novel Evoformer Decoder.

- End-to-End Differentiable Matching: Avoids NP-hard graph matching by integrating a learnable Sinkhorn-based matcher.

- Versatile Applications: Enables not only basic tasks (e.g., classification, regression) but also complex operations like graph interpolation and editing in latent space.

- SOTA Performance: Demonstrates strong results across multiple benchmarks (e.g., PUBCHEM, QM9) and outperforms baselines like PIGVAE and GraphVAE.

- Thorough Evaluation: Includes detailed ablations and empirical analysis of architecture choices, matching methods, and representation space.

- Scalability Evidence: Trained on massive molecular datasets (up to 84M samples), showing GRALE’s robustness and generalization.

Weaknesses：
- High Computational Complexity: The use of edge-level attention and Sinkhorn projections makes training and inference computationally intensive (O(N³)), limiting applicability to graphs with ≤32 nodes.

- Limited Theoretical Justification: While empirical results are compelling, theoretical guarantees (e.g., generalization bounds, robustness under graph perturbation) are not deeply explored.

- Heavy Reliance on Pretraining: The effectiveness of GRALE in downstream tasks depends strongly on massive pretraining on large datasets.

---

> ### Author Rebuttal · Authors · 2025-07-29
>
> Thank you for your detailed review. In particular, many of your questions align with research directions that we are currently investigating for future works, which we found particularly encouraging. In the following, we address the points raised in your review.
>
> > Can GRALE scale to significantly larger graphs (e.g., protein networks or social graphs with hundreds of nodes), and what would be required to do so?
>
> As you noted, the proposed model has an overall complexity of $\mathcal{O}(N^3)$. A first step toward scalability would be reducing this to $\mathcal{O}(N^2)$. While it's relatively easy to design an architecture with $\mathcal{O}(N^2)$ complexity (see baselines in Appendix E), the real bottleneck is the loss, which remains $\mathcal{O}(N^3)$ even with an oracle for the matching problem (Proposition 1). Therefore, the key would be to approximate the loss, for instance via low-rank approximations [1]. This might reduce reconstruction quality but could still yield a rich latent space.
>
> For truly large graphs (e.g., social networks), an entirely new approach would be needed. Tasks for such graphs typically focus on node representation learning. In that context, we suggest training the autoencoder to encode and reconstruct a **node’s local neighborhood**, an idea well illustrated in Figure 1 of the seminal GraphSAGE paper [2].
>
> > Would incorporating graph-specific domain priors (e.g., valency in molecular graphs) improve GRALE’s reconstructions?
>
> Yes, definitely. In this work, we deliberately avoid relying on expert knowledge to preserve generality, but incorporating such priors would certainly improve reconstruction quality. In particular, it could mitigate the artifacts discussed in Section B.3. This seems like a crucial direction for future work focused on a specific modality.
>
> > Can the Evoformer Decoder be applied independently for graph generation tasks?
>
> We believe so! The proposed Evoformer layer is defined as $A' = f(A, Z)$, where $A$ is the adjacency matrix and $Z$ is a latent variable. This makes it a promising candidate for a conditional denoising model of the form:
>
> $$A_{\text{denoised}} = f(A_\text{noisy}, \text{condition})$$
>
> > Could fine-tuning the matcher module separately improve matching accuracy for unseen graph pairs?
>
> Excellent question — it motivated us to conduct a new experiment. We fine-tuned the matcher using pairs of graphs from the training set, with 1000 gradient steps and a small learning rate. The new results (to be added to Table 5) are:
>
> **For COLORING:**
>
> | Method              | EDIT DIST | COMPUTE TIME (s) |
> |---------------------|-----------|------------------|
> | A*                  | 19.66     | 3.487            |
> | GRALE               | 08.92     | 0.005            |
> | GRALE + Finetuning  | 07.64     | 0.005            |
>
> **For PUBCHEM (note the large improvement!):**
>
> | Method              | EDIT DIST | COMPUTE TIME (s) |
> |---------------------|-----------|------------------|
> | Greedy A*           | 32.53     | 0.110            |
> | GRALE               | 32.77     | 0.008            |
> | GRALE + Finetuning  | 19.33     | 0.008            |
>
> Thanks to fine-tuning, GRALE now outperforms the best baseline by a large margin on both datasets. Thank you for this suggestion, which strengthens the paper and highlights the value of the matcher.
>
> > Is there a way to adapt GRALE to support dynamic or evolving graphs (e.g., temporal sequences)?
>
> Our preliminary experiments on graph interpolation and editing suggest that continuous dynamics in the latent space could model discrete graph-level dynamics. While this still needs to be demonstrated on more complex dynamics, it is definitely a promising avenue for future work.
>
> > How about comparing GRALE with GraphsGPT[1]?
>
> Thank you for pointing this out. GraphsGPT shares our goal of encoding graphs into Euclidean space, and we agree it merits discussion. We will update the introduction to cover methods that represent graphs as sequences (GraphsGPT and others [3,4,5]). A key limitation of these approaches is that sequencing breaks permutation invariance, which is central to many applications and to our method. Additionally, designing sequencing algorithms often requires domain-specific knowledge, reducing generality.
>
> > Scalability: The O(N³) cost for attention and matching restricts practical use to small graphs (≤32 nodes).
>
> We agree that $\mathcal{O}(N^3)$ scaling is a limitation, as stated in the conclusion. However, we would like to clarify that 32 nodes is not a strict limit. Below are time and memory costs for a single gradient step (batch size 1) on an RTX 3500 Ada (12GB VRAM):
>
> | N      | 8     | 16    | 32    | 64    | 128   | 256   |
> |--------|-------|-------|-------|-------|--------|--------|
> | Step Time (s) | 0.01  | 0.01  | 0.02  | 0.02  | 0.03   | 0.07   |
> | GPU Memory (GB) | 0.2   | 0.3   | 0.4   | 0.8   | 3.0    | 10.7   |
>
> Note that $N = 128$ fits in 3GB. This suggests that 8 A40 GPUs (48GB each) could support training with a batch size of 128 — a reasonable setting. The model used here is the same as for PUBCHEM.
>
> > Data-Hungry Training: Requires very large training sets to reach high accuracy—less suitable for low-resource domains.
>
> We implemented our modules so the model **can** leverage large pretraining datasets. With a lighter implementation, the proposed pipeline (encoder-decoder-matcher-$\mathcal{L}_{GRALE}$) could be used on smaller datasets.
>
> Since reviewer CWAG noted that "it remains unclear whether the method generalizes well to non-chemical graphs," we addressed both concerns together by training GRALE on BN, a relatively small dataset of 200,000 8-node Bayesian networks [6]. Each network has a BIC score (on the Asia dataset) used as a regression target. Using a lighter GRALE version (GNN and transformer baselines from Appendix E), we managed to outperform the state-of-the-art reported in [6]. (See reply to CWAG for detailed results.)
>
> > Generalization to Unstructured Inputs: Limited discussion or experiments on how well GRALE performs with real-world, noisy graph inputs outside curated datasets.
>
> Our experiment on COLORING (lines 252–257) suggests that GRALE is fairly robust to noise. Still, we agree this is an interesting question for real-world data and an important direction for future work, though somewhat out of scope for this paper.
>
> ### References
>
> [1] Scetbon, M., Peyré, G., & Cuturi, M. (2022). Linear-time Gromov-Wasserstein distances using low rank couplings and costs. *PMLR*.
>
> [2] Hamilton, W., Ying, Z., & Leskovec, J. (2017). Inductive representation learning on large graphs. *NeurIPS*.
>
> [3] Cofala, T., & Kramer, O. (2021). Transformers for Molecular Graph Generation. *ESANN*.
>
> [4] Xie, X. et al. (2022). From discrimination to generation: Knowledge graph completion with generative transformer. *WWW Companion*.
>
> [5] Belli, D., & Kipf, T. (2019). Image-conditioned graph generation for road network extraction. *arXiv:1910.14388*.
>
> [6] Zhang, M. et al. (2019). D-VAE: A variational autoencoder for directed acyclic graphs. *NeurIPS*.

---

> > ### Comment · Reviewer_LtMM · 2025-08-05
> > **Thanks for the rebuttal**
> >
> > Thank you for the response. I have no further questions. I recommend that the authors include a discussion in the introduction highlighting the differences between GRALE and other related works, such as GraphsGPT and references [3, 4, 5].

---

### Note · Authors · 2025-08-12

Dear reviewers and AC,

We would like to thank again the 5 reviewers for their detailed feedback. It appears that all reviewers were satisfied with the rebuttal. Therefore, we propose that our final remarks simply consist of a list of the improvements that will be added to the paper following the discussions with the reviewers.

* Fine-tuning the matcher module further improves the performances for graph matching; a new line will be added to **Table 5**. (LtMM)
* GRALE achieves SOTA performances on a new modality; this will be added to **Appendix B** "Additional experiments". (CWAG)
* We compare GRALE to an additional and strong baseline (GraphMAE); a new line will be added to **Table 3**. (CWAG)
* We report the training time of GRALE and the baselines. The table will be added to **Appendix A**. (fwJt, XSRE)
* We report the time/memory scaling of GRALE w.r.t. graph size $N$. The table will be added to **Appendix A**. (LtMM, CWAG)
* We will move Figure 4 to **Appendix B** and provide the latent structure for a wider range of quantities. (XSRE)
* We will add a discussion on methods that represent graphs as sequences in the **Introduction**. (LtMM)
* We will correct the reported typos and clarify some aspects of the model, in particular clarifying the motivations and implementation of the matcher module in **Section 3.4**. (s35a)

Our sincere thanks go to the reviewers for their thoughtful and constructive feedback, which has played a key role in refining the final version of this work.

---

### Decision · Program_Chairs · 2025-09-17

**Decision:**

Accept (poster)

**Comment:**

This paper proposes a novel graph-level autoencoder with an Evoformer-based encoder/decoder and differentiable matcher. Reviewers agree the method is technically solid, achieves strong results, and addresses concerns well in the rebuttal (including scalability, baselines, and clarity). The contributions are impactful and relevant to the community.